# Atmospheric modelling of grass pollen rupturing mechanisms for thunderstorm asthma prediction

Kathryn M. Emmerson[1]*, Jeremy D. Silver[2], Marcus Thatcher[1], Alan Wain[3], Penelope J. Jones[4], Andrew Dowdy[3], Edward J. Newbigin[5], Beau W. Picking[5], Jason Choi[6], Elizabeth Ebert[3], Tony Bannister[3]

**1** CSIRO Oceans and Atmosphere, Aspendale, Victoria, Australia, **2** School of Mathematics and Statistics, University of Melbourne, Parkville, Victoria, Australia, **3** Bureau of Meteorology, Docklands, Victoria, Australia, **4** Menzies Institute for Medical Research, University of Tasmania, Hobart, Tasmania, Australia, **5** School of BioSciences, University of Melbourne, Victoria, Australia, **6** Environmental Protection Authority Victoria, Macleod, Victoria, Australia

* kathryn.emmerson@csiro.au

**Data Availability Statement:** Measured whole grass pollen and pollen shell concentrations from Melbourne University available in S1 and S2 Tables. Radiosonde data sourced from University

## Abstract

The world's most severe thunderstorm asthma event occurred in Melbourne, Australia on 21 November 2016, coinciding with the peak of the grass pollen season. The aetiological role of thunderstorms in these events is thought to cause pollen to rupture in high humidity conditions, releasing large numbers of sub-pollen particles (SPPs) with sizes very easily inhaled deep into the lungs. The humidity hypothesis was implemented into a three-dimensional atmospheric model and driven by inputs from three meteorological models. However, the mechanism could not explain how the Melbourne event occurred as relative humidity was very low throughout the atmosphere, and most available grass pollen remained within 40 m of the surface. Our tests showed humidity induced rupturing occurred frequently at other times and would likely lead to recurrent false alarms if used in a predictive capacity. We used the model to investigate a range of other possible pollen rupturing mechanisms which could have produced high concentrations of SPPs in the atmosphere during the storm. The mechanisms studied involve mechanical friction from wind gusts, electrical build up and discharge incurred during conditions of low relative humidity, and lightning strikes. Our results suggest that these mechanisms likely operated in tandem with one another, but the lightning method was the only mechanism to generate a pattern in SPPs following the path of the storm. If humidity induced rupturing cannot explain the 2016 Melbourne event, then new targeted laboratory studies of alternative pollen rupture mechanisms would be of considerable value to help constrain the parameterisation of the pollen rupturing process.

## 1. Introduction

Thunderstorm asthma (TA) is a phenomenon characterised by a sudden increase in presentations of acute allergic asthma patients following a thunderstorm [1]. Although the mechanisms

of Wyoming: http://weather.uwyo.edu/upperair/sounding.html. Hourly PM10 and PM2.5 observations sourced from Environmental Protection Authority Victoria: https://discover.data.vic.gov.au/dataset/epa-air-watch-all-sites-air-quality-hourly-averages-yearly/historical. Code for VGPEM is available at: https://gmd.copernicus.org/articles/12/2195/2019/gmd-12-2195-2019-supplement.pdf. Radiosonde data sourced from University of Wyoming: http://weather.uwyo.edu/upperair/sounding.html.

**Funding:** This research has been supported by the Victorian Department of Health and Human Services (contract no. C5949). https://www.dhhs.vic.gov.au/ The funders had no role in study design, data collection and analysis, decision to publish, or preparation of the manuscript.

**Competing interests:** The authors have declared that no competing interests exist.

are not precisely understood, TA is thought to be caused by exposure to airborne allergenic particles such as fungal spores and pollen grains concentrated in thunderstorm downdrafts [2–4].

Grass pollen is one of the primary causes of hayfever in humans and is known to exacerbate asthma symptoms [5–7]. Whilst whole pollen concentrations correlate well with hayfever symptoms [8,9], whole pollen is too large to penetrate deep into the airways and cause TA. However, under certain conditions pollen can rupture, releasing sub-pollen particles (SPPs), which are sub-micron in diameter and capable of travelling beyond the pharynx into the small airways. SPPs have been shown to carry allergenic proteins [10,11]. It is not fully understood what causes whole pollen to rupture, but hypotheses range from ageing (fragility), mechanical friction, lightning activity within thunderstorm clouds and water-induced swelling [12,13]. The SPPs are then concentrated and transported to ground level, for example by cold downdrafts or outflows from a thunderstorm (Fig 1).

Production of SPPs via humidity induced rupturing is the most cited mechanism for causing TA. Laboratory experiments have shown pollen rupturing after submersion in water [14,15], releasing around 700 SPPs per whole grass pollen grain [16]. There is also evidence of Bermuda grass pollen rupturing in water after application of an electrical current [17], and rupturing of birch pollen after being humified for 5 hours and air dried [11]. Hughes et al. [18] were the first to determine SPPs under high rainfall conditions using single-particle fluorescence spectroscopy coincident with measurements of pollen markers such as fructose.

The world's most severe incidence of TA occurred on 21 November 2016 in Melbourne, Australia coinciding with the peak in the grass pollen season [19]. Dry storms with an associated gust front swept eastwards through Melbourne at 17:30 local time (06:30 UTC), when many people were outdoors. The 2016 Melbourne event was preceded by very hot dry weather (maximum temperature of 34°C), with strong north-westerly winds capable of transporting grass pollen and dust from surrounding agricultural regions to populated areas. Scant rainfall fell after the gust front passed through, with up to 4 mm west of Melbourne and up to 2 mm in the east [20]. Grundstein et al's. [21] analysis of downdraft convective available potential energy over Melbourne was strongly suggestive of downdraft activity. The line of easterly moving storms that developed on 21 November meant that the strongest downdraft outflow pushed forwards in an easterly direction.

Large volumes of calls to the emergency services reporting severe breathing difficulties started occurring from 07:00 UTC after the storm front passed, suggesting that exposure to allergenic particles borne by the storm occurred immediately [19]. During that night, there was a 672% increase in patients arriving at emergency departments with respiratory problems [20], eventuating in 10 fatalities. The Melbourne event was extremely challenging to health service providers due to the sudden and unforeseen influx of large numbers of patients. One of the challenges in preparing for such episodes is their relative rarity; whilst south-eastern Australia and Melbourne in particular have the highest frequency of TA episodes globally [12], Silver et al. [22] estimate TA occurs in Melbourne approximately only once every five years.

The Victorian Grass Pollen Emission Module (VGPEM) [23] was developed to predict the risk of TA occurring as part of the Australian Bureau of Meteorology's pilot forecasting system [24]. VGPEM is based on the aerodynamic properties (diameter and density) of ryegrass pollen (*Lollium perenne*). Ryegrass is an important pasture grass grown in large regions of western Victoria [25], and has very strong proven allergenic properties [26]. Ryegrass flowers in the austral spring, releasing pollen with a diameter of 30–40 μm. Prior to the 2016 pollen season, a wetter than average winter contributed to the vigorous growth of ryegrass [20], leading to a prediction that the season ahead would be problematic for sufferers [27]. During the 2017 grass pollen season in Victoria (October to December) VGPEM predicted the correct grass

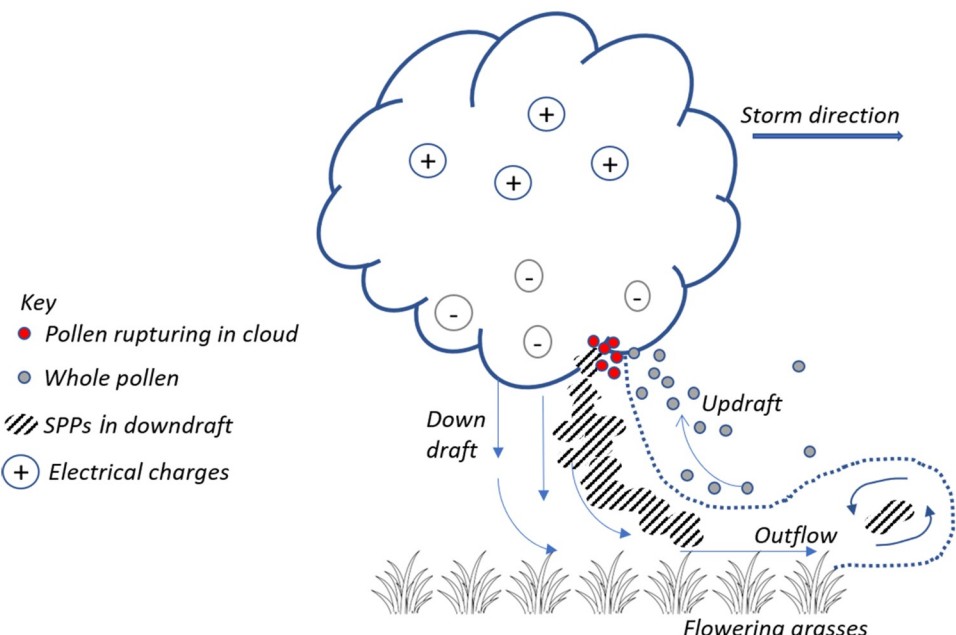

**Fig 1. Schematic of hypothesised pollen entrainment into a thunder cloud, rupturing and deposition mechanisms/processes.** After Taylor and Jonsson [13].

pollen category [28] on average 69% of the time across eight sites in the newly set-up Victorian Thunderstorm Asthma Pollen Surveillance (VicTAPS) network.

To date, the role of humidity induced grass pollen rupturing in TA has not been explored via atmospheric modelling studies. In this paper, we further develop VGPEM to test humidity induced pollen rupturing as a mechanism for the 2016 Melbourne event. We also use the model to test several alternative meteorological-driven mechanisms able to produce SPPs in the Melbourne atmosphere at the time of the storm, using inputs from three different meteorological models. We also investigate whether a dust parameterisation can produce a correctly timed increase in particulate matter, which provides confidence in the meteorological modelling.

## 2. Methods

### 2.1 Observations of pollen

For the purposes of this study, we obtained hourly average grass pollen observations using standard methods for a 34-hour period beginning on 20 November at 04:30 UTC and ending on 21 November at 13:30 UTC. The grass pollen measurements derive from a Hirst-type pollen trap (Burkard Manufacturing) situated on a roof-top at Melbourne University (144.965˚E -37.797˚N, 43 m elevation). Pollen is collected on slides coated with Sylgard adhesive, then prepared with Calberla's stain to make observation easier and counted using light microscopy. The number of whole grass pollen and broken or empty pollen grains (called pollen 'shells' hereafter) was assessed across each hourly transect. We could not distinguish with confidence the taxa of these pollen shells; thus a total count is presented. Usual practice is to count pollen across 24-hour transects, as the process is manually demanding; for this study we counted vertical transects to obtain hourly measurements around the 21 November event, with pollen counts converted into grains m$^{-3}$ (S1 Table in S1 File). 24-hour pollen counts on either side of

21 November were too low to consider hourly subdivision (S2 Table in S1 File). The respirable SPPs are too small to impact on the volumetric spore sampler and are not detectable with standard light microscopy procedures.

## 2.2 The Victorian Grass Pollen Emissions Model

The Victorian Grass Pollen Emissions Module (VGPEM1.0) treats whole pollen as 35 μm diameter particles with a density of 1000 kg m$^{-3}$ [23]. The model uses the satellite derived Enhanced Vegetation Index product to estimate the seasonal variation in grass pollen emissions, together with hourly meteorological inputs for the shorter-term variations. High temperatures and low relative humidity (RH) promote pollen emission. The horizontal resolution of VGPEM is 3 km across Victoria, using 306 x 306 grid cells. The temporal resolution is hourly, time-stamped at the beginning of the hour.

VGPEM is an emissions module inside the CSIRO Chemical Transport Model (C-CTM) [29], which provides the atmospheric dispersion and deposition capabilities. In this work we drive the C-CTM with meteorological forecasts from three different weather models to gain an ensemble view representing uncertainty in the meteorological conditions. The first model is the numerical weather prediction version of the Australian Community Climate and Earth System Simulator (ACCESS) [30]. The ACCESS domain used here is nested within a global ACCESS run and does not require separate boundary conditions. The second model is the Conformal Cubic Atmospheric Model (CCAM) [31] using boundary conditions from ERA-Interim [32]. The third model version 4.1.1 of the Weather Research and Forecasting model (WRF) [33] using boundary conditions from National Centers for Environmental Prediction Final reanalyses (NCEP FNL) [34]. Details of the meteorological models used are given in S3 Table in S1 File. CCAM and WRF meteorological capabilities were evaluated across south east Australia and found to reproduce temperatures and local scale meteorological features well, with wind speeds and water vapour mixing ratios within benchmark ranges [35].

The C-CTM has previously been used with both ACCESS and CCAM and has an interface designed for both inputs. The vertical dimensions of ACCESS and CCAM are structured differently; ACCESS uses hybrid model levels (to 29 km on 67 levels), whilst CCAM is output on pressure levels (to 2.2 hPa on 35 levels). WRF is not interfaced with the C-CTM but is similarly structured on pressure levels like CCAM and was therefore reformatted to be used with the CCAM input interface. The C-CTM resolves to 11 km for ACCESS and 444 hPa in CCAM/ WRF (~6 km). For consistency, data are stored for all models to a height of ~5 km.

In this paper we refer to the meteorological results from each of the driving models by their names, e.g. ACCESS, CCAM and WRF. The different results of whole pollen, pollen shells and SPPs predicted by the C-CTM and driven by each of these meteorological models will be referred to as ACCESS$^{\text{C-CTM}}$, CCAM$^{\text{C-CTM}}$ and WRF$^{\text{C-CTM}}$.

## 2.3 In-atmosphere humidity induced grass pollen rupturing process

We represent grass pollen rupturing following the theoretical approach of Wozniak et al. [36]. They describe in-atmosphere rupturing, which takes place using an 80% RH threshold. Wozniak et al. [36] treat this rupturing mechanism as an emission, whereas in this work we treat the in-atmosphere rupturing as an instantaneous process depending on the concentration of whole grass pollen grains present in any grid cell (g m$^{-3}$). Note here that the C-CTM carries pollen in the model as a mass concentration of aerosol, which is converted to grains m$^{-3}$ using the mass of one ryegrass pollen grain, $m_{pol} = 22.4 \times 10^{-9}$ g derived in Emmerson et al. [23]. Once the pollen concentration is converted to grains m$^{-3}$, $\chi$, we calculate the number of whole grass pollen grains ruptured, $N_{rupt}$ (grains m$^{-3}$) if at least one pollen grain per m$^3$ is present (1

grain m$^{-3}$ = 0.0224 µg m$^{-3}$) and when the RH reaches 80% at any model grid cell in space and time:

$$N_{rupt} = F_{rupt} \times \chi, \tag{1}$$

where $F_{rupt}$ is the fraction of whole grass pollen grains that rupture (= 0.7), based on the fraction of ryegrass pollen grains that rupture in water after 5 minutes [15]. $F_{rupt}$ is the same for all simulations in the absence of literature on other rates of ryegrass pollen rupturing. We assume that 1 empty shell is produced per whole pollen grain. Using $N_{rupt}$ we calculate the number of SPPs generated by multiplying by the number of SPPs produced per grain of pollen, $n_{spg}$. The measurements of Suphioglu et al. [16] on ryegrass showed around 700 SPPs per whole pollen grain of a size range 600 nm– 2.5 µm. We will use 700 SPPs per whole pollen grain with an SPP diameter of 600 nm which is the lower end of Suphioglu et al's [16] measurements. The SPPs have the same density as whole grass pollen. This yields a mass of one SPP, $m_{SPP}$ = $1.13 \times 10^{-13}$ g and a fall speed of 0.01 cm s$^{-1}$. Upon rupturing, we convert the number of SPPs generated per m$^3$ back into a mass concentration (using $m_{SPP}$) to be carried by the C-CTM. This mass concentration is used twice; to be lost from the tracer containing the mass of whole pollen, and to be added to a tracer containing the mass of SPPs. We also track the number of whole pollen grains that rupture ($N_{rupt}$), as there are no observations of SPPs, only the pollen shells.

## 2.4 Humidity induced on-plant mechanical rupturing

We also include Wozniak et al's [36] on-plant mechanical rupturing, $M_{rupt}$ of pollen on the plant, which occurs when exposed to moisture from humidity and precipitation [11,15]. Similar to Wozniak et al. [36], we treat mechanical rupturing as an emission occurring at the surface only (g m$^{-2}$ s$^{-1}$):

$$M_{rupt} = F_{rupt} n_{spg} P_{fx} (1 - f_{PR} f_{RH}) \frac{m_{SPP}}{m_{pol}}, \tag{2}$$

where $P_{fx}$ is the emission rate of grass pollen from the plant (g m$^{-2}$ s$^{-1}$) predicted by VGPEM, and $m_{SPP}/m_{pol}$ is the mass ratio of SPPs to whole grass pollen grains. The terms $f_{PR}$ and $f_{RH}$ are the emission activity factors for precipitation and RH, respectively. The idea is that mechanical rupturing will occur on the fraction of grass pollen left on the plant when humidity conditions are too high to warrant direct pollen emission to the atmosphere. The emission activity factors are calculated in the model using a logistic function, $f_l$ of the form:

$$f_l(y : \alpha, c) = \frac{1}{1 + e^{\alpha(y-c)}}, \tag{3}$$

Where y is the variable of interest, e.g. RH or precipitation, for rate parameter $\alpha$ and location parameter c. For RH, the function follows Sofiev et al. [37]:

$$f_{RH} = f_{baseline} + (1 - f_{baseline}) \times f_l(RH; \alpha_{RH}, c_{RH}) \tag{4}$$

The rate and location parameters will give $f_l$ = 0.95 at 50% RH and 0.05 at 80%. $f_{baseline}$ represents the fraction of pollen emitted at very low RH (= 0.33) and was used in Emmerson et al. [23].

The precipitation function $f_{PR}$:

$$f_{PR} = f_{baseline} + (1 - f_{baseline}) \times f_l(PR; \alpha_{PR}, c_{PR})/f_l(0; \alpha_{PR}, c_{PR}), \tag{5}$$

where the logistical function parameters result in values of 0.95 for no precipitation falling and 0.05 for precipitation at 0.5 mm h$^{-1}$. The terms $\alpha_{RH}$ and $\alpha_{PR}$ are both negative, meaning the pollen emission decreases as either the humidity or precipitation rates increase.

## 2.5 Using strong wind gusts as a mechanism for in-atmosphere rupturing

Given the strength of winds in the storm gust front it seems likely that a wind driven rupturing process could represent the pollen shell observations. Visez et al. [38] simulated the action of wind blowing birch pollen against solid surfaces and found higher concentrations of SPPs associated with stronger wind speeds. Here we test two variations; one using a function of wind speed, $f_{WS}$ described in Sofiev et al. [37], the other using wind speed (WS) directly to rupture the pollen. The function of wind speed takes the form:

$$f_{WS} = f_{baseline} + (1 - f_{baseline}) \times (1 - exp(-WS/U_{sat})), \tag{6}$$

where $U_{sat}$ is a saturation wind speed of 5 m s$^{-1}$ which constrains the pollen emission, and $f_{baseline}$ is 0.33 as before. The $U_{sat}$ constraint suggests that high winds can only promote pollen release if pollen is available on the plant. The function takes the form of a curve, beginning at 0.33 for zero wind speed and asymptotes to 1 at wind speeds >15 m s$^{-1}$. The pollen rupturing occurs using a variation to Eq 1 with no threshold value:

$$N_{rupt} = F_{rupt} \times \chi \times f_{WS}, \tag{7}$$

In the second case of rupturing directly with wind speed, we ensure windspeeds are above a threshold of $U_{sat}$. This ensures that rupturing only occurs at higher wind speeds, as $f_{WS}$ will allow rupturing even at negligible wind speeds. The threshold $U_{sat}$ value of 5 m s$^{-1}$ is equivalent to $f_{WS}$ = 0.75. Grass pollen rupturing then occurs similar to Eq 1, but with the condition of RH replaced with the $U_{sat}$ threshold.

## 2.6 On-plant mechanical rupturing using wind gusts

A prerequisite of in-atmosphere rupture is the presence of airborne pollen that is already disperse. We adjust Wozniak et al's description of mechanical rupturing from Eq 2 to represent a wind speed induced emission of 600 nm particles from the ground, replacing $(1-f_{PR}f_{RH})$ with either $f_{WS}$ or wind speed as the part of the definition, e.g.:

$$M_{rupt} = F_{rupt} n_{spg} P_{fx} f_{WS} \frac{m_{SPP}}{m_{pol}}, \tag{8}$$

This process considers the resuspension of previously deposited SPPs.

## 2.7 Electrical charge associated with low relative humidity

When air is humid, the moisture aids the transfer of electrical charges between atoms. At lower RH the electrical transfer is less effective and higher charges can build up. Vaidyanathan et al. [17] showed that electrical activity could trigger release of allergenic material, and observed Bermuda grass pollen rupture occurring when subjected to voltages of 10 kV m$^{-1}$. Taylor et al. [39] thought the same was likely of ryegrass pollen, given its high allergenic properties and association with thunderstorm asthma, even in air with an RH of 40%. We explore the hypothesis that pollen particles could become electrically charged when the RH is low. In this simulation, we use Eq 1 again to represent this process but use a threshold RH value of $\leq$ 30% to start pollen rupturing.

## 2.8 Lightning

The presence of lightning strikes has been associated with increased asthma presentations generally [40], but not necessarily in conjunction with an epidemic event.

Lightning is not simulated by any of our meteorological models and are incorporated into the pollen simulations from lightning observations. The number of lightning occurrences per hour, $n_{lightning}$, was re-gridded to our 3km resolution based on data from the World Wide Lightning Location Network (WWLLN) comprising of observations from a ground-based network of sensors [41]. The WWLLN includes both cloud-to-ground and cloud-to-cloud lightning, but preferentially detect the stronger cloud-to-ground occurrences as noted by Virts et al. [42]. If lightning occurred in any grid cell where whole pollen grains were also present, then this was used as the rupturing trigger. Eq 1 was again modified to include the lightning count, meaning that rupturing was stronger if there were more lightning occurrences:

$$N_{rupt} = F_{rupt} \times \chi \times n_{lightning},$$ (9)

## 2.9 Dust

We examined existing C-CTM tracers representing mineral dust to replicate the action of particles being swept into the atmosphere by the strong winds in the storm inflow and outflow. The dust parameterisation relies on individual particles being saltated from land designated as bare ground and having a low soil moisture [43]. Three dust tracers are summed to represent $PM_{10}$ (<2.5, 2.5–5 and 5–10 μm diameter).

In total, VGPEM has been set up to run with eight pollen rupturing mechanisms, and the C-CTM dust tracer described above. The different rupturing thresholds and rupturing equations are compared below in Table 1. In each run, the model will carry whole pollen, pollen shells and SPPs.

## 3 Results and discussion

First, we examine the time-series of observed whole grass pollen at Melbourne University, which shows a maximum hourly average of 177 grains m$^{-3}$ the day before the storm (Fig 2a), and lower concentrations on the day of the storm. Note the 24-hour average grass pollen measurement dated 21 November was 102 grains m$^{-3}$, taken at 4pm local time, and therefore includes the grass pollen deposited on the trap after the slide was changed on 20 November at 4pm. The models also predict higher whole pollen concentrations the day before the storm but get the timing of this peak too early. At the time of the storm (06:30 UTC on 21 November) all the models capture the correct magnitude of whole grass pollen in the surface layer of the atmosphere. The observed whole grass pollen peaks at 131 grains m$^{-3}$ at 05:30 UTC before dropping sharply to 40 grains m$^{-3}$ by 08:30 UTC after the storm. The models also predict more than 100 grains m$^{-3}$ in the hours prior to the storm, but the sharp decline occurs one hour later than the observations. Despite the mismatch in timing, the models show whole pollen is available in the atmosphere to be ruptured. Getting the magnitude in whole grass pollen grains correct is an important constraint for the pollen rupturing process, ensuring that neither too little nor too much whole pollen will be ruptured. However, our evaluations are limited by only one set of observations in the model domain (2016 was prior to the existence of the VicTAPs network).

We do not have measured SPP concentrations in this work. However, the observed concentration of pollen shells might indicate the potential concentration of SPPs present in the

**Table 1. Thresholds and variables used in each of the model experiments.**

| Experiment name | Description | In-atmosphere rupture, $N_{rupt}$ | On-plant mechanical rupture, $M_{rupt}$ |
|---|---|---|---|
| RH $\geq$ 80% | Humidity induced rupturing when RH is $\geq$ 80% | $F_{rupt} \times \chi$ | |
| $M_{rupt}$ RH $\geq$ 80% | | $F_{rupt} \times \chi$ | $F_{rupt} n_{spg} P_{fx} (1 - f_{PR} f_{RH}) \frac{m_{SPP}}{m_{pol}}$ |
| $f_{WS}$ | Rupturing (or emission) using a function of wind speed | $F_{rupt} \times \chi \times f_{WS}$ | |
| $M_{rupt}$ $f_{WS}$ | | $F_{rupt} \times \chi \times f_{WS}$ | $F_{rupt} n_{spg} P_{fx} f_{WS} \frac{m_{SPP}}{m_{pol}}$ |
| WS $\geq$ 5 m s$^{-1}$ | Rupturing (or emission) using wind speeds $\geq$ 5 m s$^{-1}$ | $F_{rupt} \times \chi \times WS$ | |
| $M_{rupt}$ WS | | $F_{rupt} \times \chi \times WS$ | $F_{rupt} n_{spg} P_{fx} WS \frac{m_{SPP}}{m_{pol}}$ |
| RH $\leq$ 30% | Static electricity induced rupturing when RH is $\leq$ 30% | $F_{rupt} \times \chi$ | |
| Lightning | Lightning induced rupturing | $F_{rupt} \times \chi \times n_{lighting}$ | |
| Dust | Uses a dust tracer | Not applicable | |

$F_{rupt}$ = fraction of whole pollen shells that rupture = 0.7. $\chi$ = airborne concentration of whole pollen, µg m$^{-3}$, $n_{spg}$ = number of SPP per whole pollen grain = 700. $P_{fx}$ = whole pollen emission rate, g m$^{-3}$. $F_{pr}$ = precipitation function. $f_{RH}$ relative humidity function. $M_{SPP}$ = mass of 1 SPP = 1.13 x 10$^{-13}$ g. $M_{POL}$ = mass of 1 whole pollen grain = 22.4 x 10$^{-9}$ g. $f_{WS}$ = function of wind speed. $WS$ = wind speed, m s$^{-1}$. $N_{lightning}$ = number of lightning occurrences.

atmosphere (Fig 2b). Over the 34 hours of observations, the average concentration of pollen shells is approximately 68% of the concentration of whole grass pollen grains, with a range of 5 to 138%. At the time of the thunderstorm, the concentration of pollen shells was 48 grains m$^{-3}$ or 51% of the whole grass pollen grains. This is only approximately one third of the peak recorded over the 34-hour observation period. The peak in pollen shell concentrations (134 grains m$^{-3}$) was measured at 10:30 UTC on 20 November, 20 hours before the storm. Thus there is a lack of correlation between pollen shell concentrations and reported asthma events. This lack of correlation was also noted by Buters et al. [44] and Plaza et al. [45], who suggested that the allergenic properties of SPPs could vary 10-fold depending on how ripe the pollen was at release time. This produces other questions: could the SPPs have been more potent on 21 November as opposed to 20 November? Can whole pollen grains completely shatter and not leave a pollen shell behind (as opposed to only ejecting SPPs though the germination pore)? Could whole pollen also rupture within the windpipe due to moist conditions, similar to Sporik et al's [46] research on fungal spores germinating in the respiratory tract and causing inflammation? The lack of correlation might also suggest that other fine particle sources

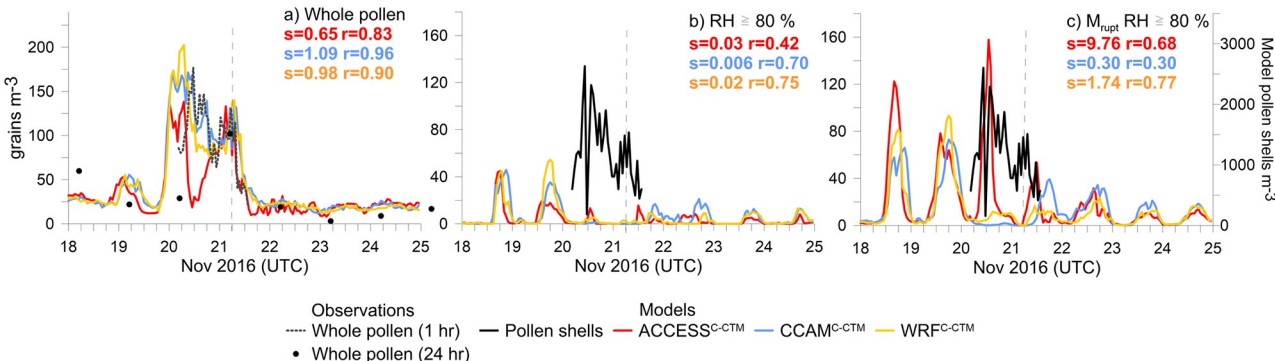

**Fig 2. Time series in whole grass pollen (a) and pollen shells (b–c) from the RH $\geq$ 80% experiment.** Panel (b) shows results from in-atmosphere only pollen rupturing, whilst panel (c) includes the on-plant mechanical rupturing from Eqs 2–5. Model values in (c) are on second y-axes to the right. s = slope of the linear regression and r = r correlations between modelled and observed pollen (or pollen shells), given in same colours as the legend. Vertical dashed line indicates the time of the storm.

contributed to human health effects, such as fungal spores [2,4]. Marks et al. [3] did show increases in both pollen shells and whole grass pollen during a 1997 storm in New South Wales, but the ratio of pollen shells to whole pollen did not increase due to the storm. By contrast, Hughes et al. [18] observed a decrease in whole pollen concentrations due to the rain.

## 3.1 Humidity induced grass pollen rupturing

Using the hypothesised rupturing process with an 80% RH threshold does not yield a sharp rise in the modelled pollen shells (or the SPPs) at the time of the storm front, as we would expect to observe given the huge increase in respiratory problems (Fig 2b). Key to the Hughes et al. [18] finding was that SPP increases were associated with high rates of rainfall coincident with thunderstorm activity. The difference in the Melbourne storm was the absence of rain. Lack of rain is the major factor in why so many people were outside at the time and were exposed. The models show that humidity induced rupturing was strongest at other times during the week e.g. on 18 and 19 November (Fig 2b), and also tends to occur each night rather than during the day (due to lower temperatures). Inclusion of the on-plant mechanical rupturing (Fig 2c) adds thousands of pollen shells into the atmosphere, but the process reinforces the wrong timing of the rupturing (at night) rather than contributing to the TA event. Using the RH ≥80% rupturing mechanism in the models under-predicts the pollen shell concentrations by a factor of ~30 for the in-atmosphere only case, whilst inclusion of the mechanical rupturing overpredicts by a factor of ~80.

Prior to the arrival of the gust front, observations from the closest surface weather station to Melbourne University (Olympic Park) showed very low RH of 18% (Fig 3a). Three other weather stations upwind of the airflow to the north (Essendon and Melbourne Airports) and west (Laverton) also observed RH between 18–19%. Humidity induced rupturing at the surface on the day of the storm could only occur when RH reached 80% some five hours after the storm had passed.

Differences in the timing of observed and predicted grass pollen concentrations could be due to errors in the modelled meteorological forcing. Of the meteorological models driving VGPEM, ACCESS and WRF capture the timing of the humidity increase and associated sudden temperature decrease (Fig 3b) after the storm passes better than CCAM, though wind speeds predicted by all models are lower than was observed prior to the storm (Fig 3c). Both CCAM and WRF tend to predict stronger wind speeds than ACCESS, and only CCAM predicts the correct wind speed at the time of the storm. All models predict RH above 80% at night on 19 and 21 November, whilst ACCESS also has a high RH at night on 20 November, coinciding with the timing of rupturing in Fig 2b and 2c.

Radiosonde soundings from Melbourne Airport (20 km north of central Melbourne) show the vertical profiles in RH at 00:00 UTC and 12:00 UTC; six hours before and after the storm (Fig 4a and 4c, respectively). None of the models track the observations precisely, and the shape of the modelled RH profiles tend to be better after the storm at 12:00 UTC. Vertical profiles of RH in the models are shown at 06:00 UTC (Fig 4b) and all show an increase in RH around 600 hPa which is where we expect the storm clouds to be. The high observed humidity at 600 hPa ±6 hours either side of the storm could reasonably be expected to persist throughout the day, so we infer the predictions of the meteorology models are reasonable at 06:00 UTC.

We plot the 80% RH threshold in Fig 4b at the time of the storm, showing only ACCESS and WRF reach the required 80% RH at around 600 hPa (Fig 4b). In Fig 4d the corresponding vertical profile in whole grass pollen at Melbourne Airport shows whole grass pollen grains present throughout the lowest 5 km of the modelled atmosphere. The 1 grain $m^{-3}$ threshold is lost at ~800 hPa in ACCESS$^{C-CTM}$ and CCAM$^{C-CTM}$ or ~2 km, and 630 hPa in WRF$^{C-CTM}$.

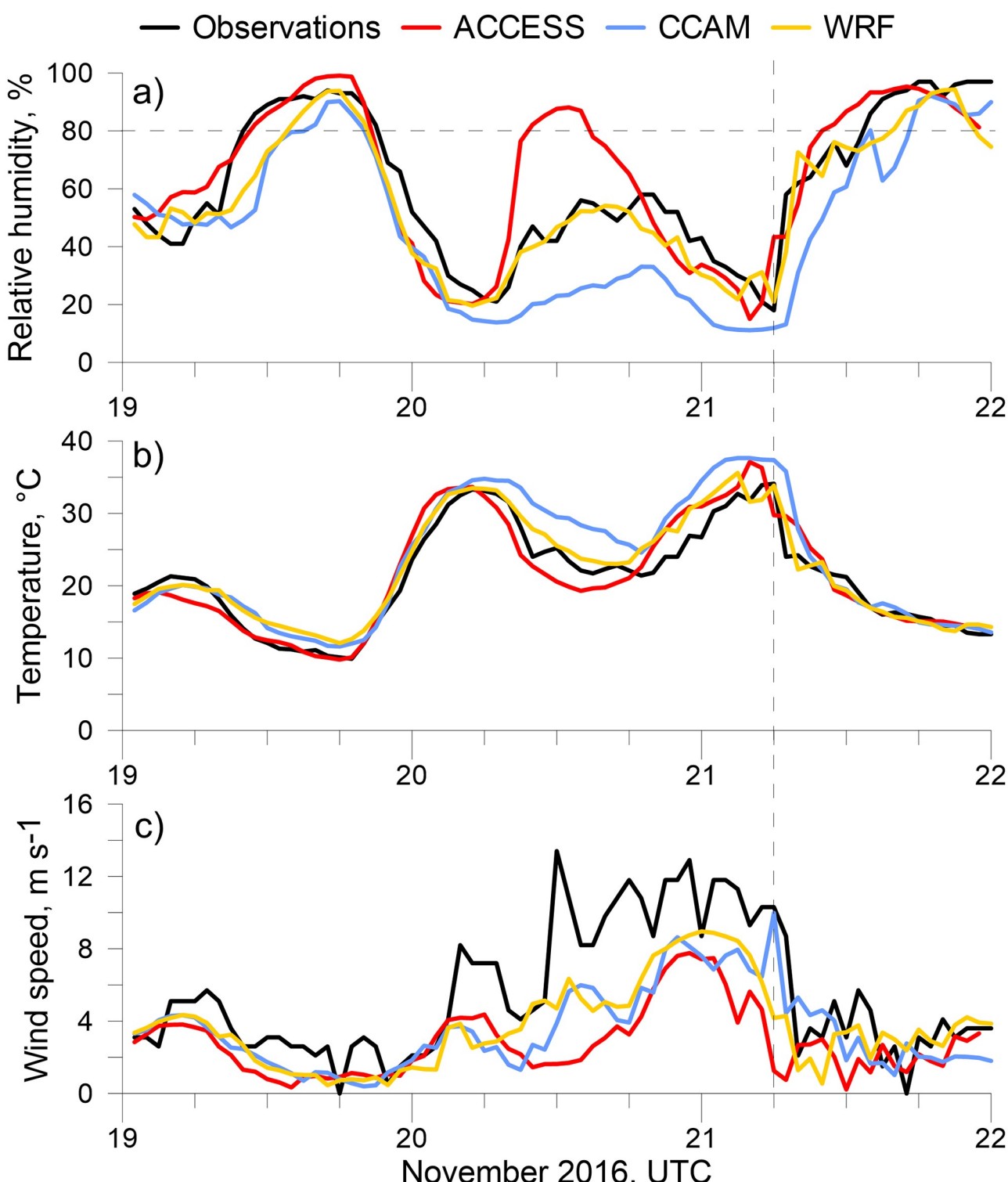

**Fig 3. Time series in (a) 2m relative humidity (b) 2 m temperature and (c) 10 m wind speed at Melbourne Olympic Park, the closest automatic weather station to the pollen observation site.** The vertical dashed line indicates the time of the storm, the horizontal dashed line indicates the 80% RH threshold.

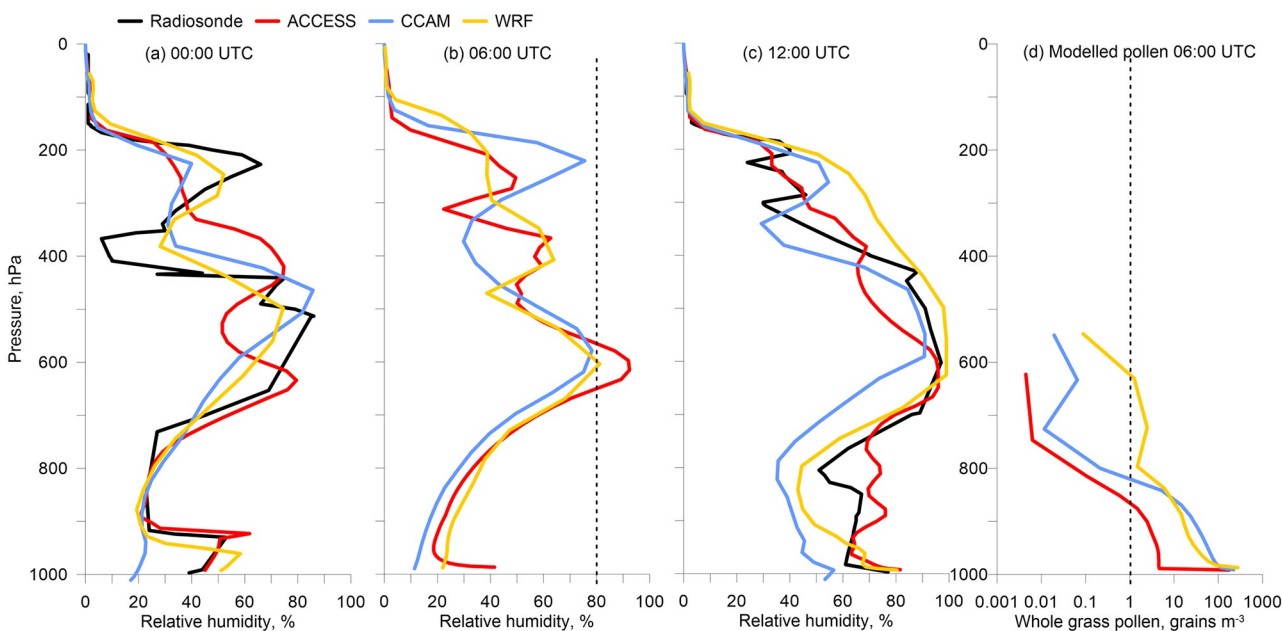

**Fig 4.** (a-c) radiosonde observations at Melbourne Airport compared to model predictions for the 21 November 2016 of relative humidity from the surface to 0 hPa in each meteorological model. (d) pollen concentrations in VGPEM1.0 driven by each meteorological model. Dashed lines in (b) and (d) represent the thresholds for rupturing at 80% RH and 1 grain m$^{-3}$, respectively. The maximum height of modelled pollen differs because CCAM and WRF are output on pressure levels, whilst the ACCESS height is converted to hPa from the 5 km model level.

Vertical concentrations of whole grass pollen decrease exponentially from the surface, with most (62% average across models) situated in the first 40 m. It is only in the WRF$^{C-CTM}$ profile that the whole pollen may intersect with RH conditions being high enough for rupture, noting that observations plotted in Fig 4 were above Melbourne Airport whereas rupturing may have occurred at other locations in the model domain.

Large differences exist between the models in the whole pollen concentrations produced with altitude and highlights how running an ensemble is beneficial. Whilst our results, and those from Wozniak et al. [36] show an exponential decline in whole pollen concentrations with altitude, measurements at 2000m altitude in the mountainous Thessaloniki region of Greece show grass pollen 2.6 times less than the surface concentrations [47]. Our models produce whole pollen at 2000 m in the range 185–1770 times less than the surface, whereas the Wozniak et al [36] model produces >500 times less in the north east region of the USA in spring. We tested whether artificially increasing the model vertical whole pollen concentration in the RH ≥ 80% experiment would impact the rate of rupturing at the time of the storm. We applied a linear relationship such that the ratio of surface to 2000 m whole pollen concentrations is 2.6:1. The results are largely similar, with the RH ≥80% threshold producing more pollen shells and SPPs in the models outside of the storm period (and mainly at night) using this mechanism (S1 Fig in S1 File).

The vertical time series to 5 km from each model at Melbourne University shows whole grass pollen concentrations are maintained for some time before and after the storm (Fig 5a–5c), indicating availability of pollen to be ruptured. We still see most of the whole pollen concentrated close to the surface. There are very low concentrations of SPPs throughout the ACCESS$^{C-CTM}$ and CCAM$^{C-CTM}$ 5 km model atmospheres before and at the time of the storm (Fig 5d and 5e). The pre-gust front air was well mixed and turbulent, sweeping SPPs out of the

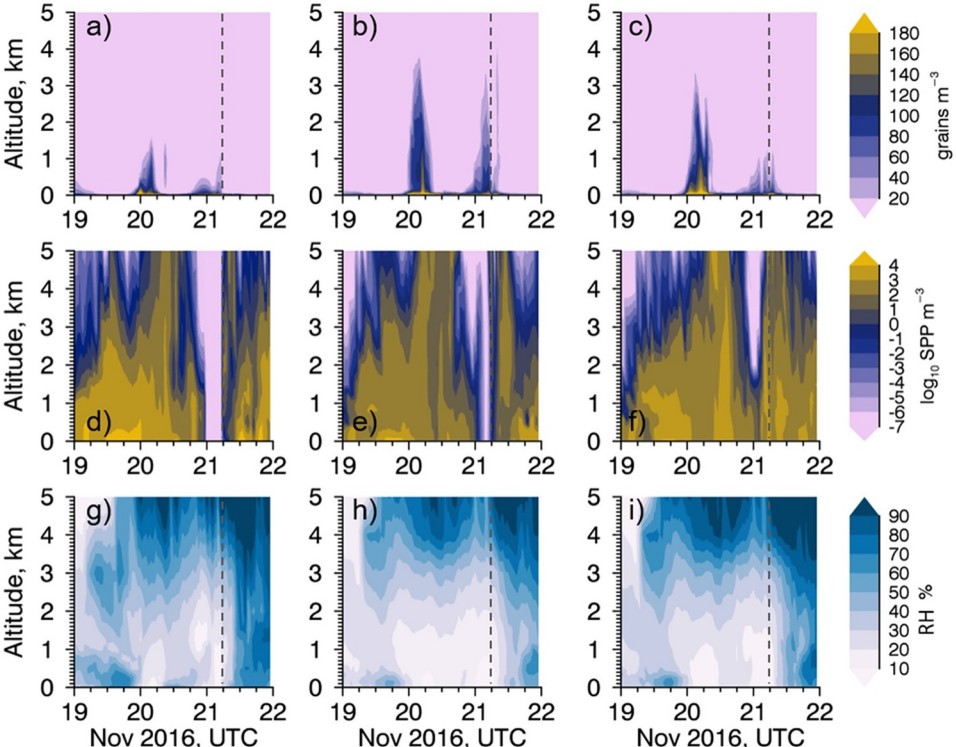

**Fig 5. Time series with altitude plots of whole grass pollen (a–c), SPPs to log base 10 (d–f), and relative humidity (g–i) in the atmosphere above the pollen counting site in Melbourne.** ACCESS[C-CTM] is shown in the left hand panels (a,d,g), CCAM[C-CTM] in the middle panels (b,e,h) and WRF[C-CTM] in the right hand panels (c,f,i). Vertical dashed line indicates the time of the storm.

Melbourne air, with no opportunity for resupply as the RH levels were between 10–30% to a height of 2 km in each of the models. The WRF[C-CTM] model shows SPPs present at the time of the storm, though this only translates into a pollen shell concentration of 2.2 grains m$^{-3}$ at the surface (Fig 2b). Modelled pollen shell concentrations were higher on 19 and 20 November, but these dates did not coincide with health impacts.

To summarise our initial findings, we find that the RH throughout the atmosphere at central Melbourne to be well below the 80% required for pollen rupturing at the time of the storm on 21 November. The key finding is that more SPPs are produced regularly at other times when RH tends to be higher, such as at night rather than during a hot late spring day. The high risk of false positives does not make the RH $\geq$80% mechanism a good one for the prediction of thunderstorm asthma. On the day of the storm, SPPs were only produced in the models some hours after the storm had passed, as the humidity increased. This is too late to have caused the spike in emergency calls at 07:00 UTC in Melbourne. The condition of the 80% RH threshold required for pollen rupturing is also a condition that suppresses whole pollen emissions from the plant in VGPEM via the closing of anthers at high humidity. In the laboratory rupturing experiments cited earlier, pollen was observed to rupture after full submersion in water, or humidified for hours, and it is not clear whether a brief intersection of pollen and high humidity is enough to cause rupturing in the atmosphere.

## 3.2 Alternative mechanisms for the 2016 Melbourne event

We know that a process occurred in the atmosphere at 06:30 UTC on 21 November 2016, causing a high concentration of respirable particles to be inhaled. In this section we focus on the results of the other model experiments that could possibly explain how a sudden source of SPPs might have occurred. Each of the panels (b-g) in Fig 6 shows each experiment producing a high concentration of pollen shells, some of which produce peaks in line with the pollen shell observations (highest on 20 November, gives higher correlations e.g. (r = 0.86–0.91 in Fig 6d), and some which produce higher peaks at the time of the storm (with corresponding poor correlations, e.g. (r = 0.48–0.54 in Fig 6g).

Fig 6a is different and shows the time series in the model dust tracers compared to hourly $PM_{10}$ measurements from Footscray, the closest air quality monitoring site to the pollen observations. The dust tracers are normalised to the observed peak value for visual comparison, as dust only represents one fraction of total $PM_{10}$. This experiment shows the three meteorological models simulate the gusty winds well, and that wind-driven processes would cause particles to spike in the Melbourne atmosphere during the storm. The timing of the modelled $CCAM^{C-CTM}$ and $WRF^{C-CTM}$ dust peaks coincides with the peak in hourly $PM_{10}$ measurements (Fig 6a), with $ACCESS^{C-CTM}$ peaking slightly later. The peak of 699 µg m$^{-3}$ in the 5-minute observations (not shown) occurred at 06:55 UTC as the storm front passed and $PM_{10}$ remained above 100 µg m$^{-3}$ for 20 minutes. There was no increase in observed $PM_{2.5}$ mass concentrations across Melbourne air quality stations (at Footscray, Alphington and Geelong South), although whole pollen and dust are generally found in more coarse size fractions. Assuming SPPs were responsible for the 2016 Melbourne event, the $PM_{2.5}$ observations at the three Melbourne stations suggest SPPs were too small to contribute much mass to $PM_{2.5}$.

We now investigate wind driven rupturing processes. The time series in modelled and observed pollen shells fit quite well using both $f_{WS}$ and WS parameterisations (Fig 6b and 6c), suggesting more whole pollen ruptured on the evening of 20 November. However, $ACCESS^{C-CTM}$ predicts a slightly higher peak in pollen shells at the time of the storm, whilst $CCAM^{C-CTM}$ and $WRF^{C-CTM}$ predict more pollen shells the day before. More pollen shells result from $f_{WS}$ than the WS experiment because rupturing occurs without a threshold.

On-plant mechanical rupturing using $f_{WS}$ produces the best correlations between the model and observed pollen shells of all our experiments (r = 0.86–0.91 in Fig 6d). Whilst on-plant mechanical rupturing using $f_{WS}$ might therefore provide the best parameterisation of pollen shell concentrations in the atmosphere, it does not explain the severe respiratory impacts of the storm. On-plant mechanical rupturing using WS directly with a threshold of 5 m s$^{-1}$ (Fig 6e) produces a peak in pollen shells at the time of the storm in all three models ($CCAM^{C-CTM}$ also produces a larger peak a few hours prior). The vertical time series in wind speed at Melbourne (not shown) shows wind speeds of up to 18 m s$^{-1}$ throughout the lowest 5 km of the atmosphere, starting at 12:00 UTC on 20 November and remaining high above 2 km throughout 21 November. The downward transport of horizontal momentum in the thunderstorm downdraft likely contributed to the gusty winds that occurred on 21 November.

As $f_{WS}$ yields a value between 0 and 1 and the wind speed can be any value, the pollen shell concentrations resulting from the WS on-plant mechanical rupturing experiment are much higher (Fig 6d and 6e). In terms of the ratio of in-atmosphere to on-plant mechanical rupturing, mechanical rupturing provides 95–96% of the total concentration of pollen shells comparing Fig 6 panels (b) and (d) for $f_{WS}$ across all models and 80–94% comparing Fig 6 panels (c) and (e) for WS. Note here that as mechanical rupturing takes place before the pollen leaves the plant, the threshold of whole pollen grains in the air does not apply. Therefore, there are many thousands of pollen shells resulting from the mechanical rupturing experiments, as there is

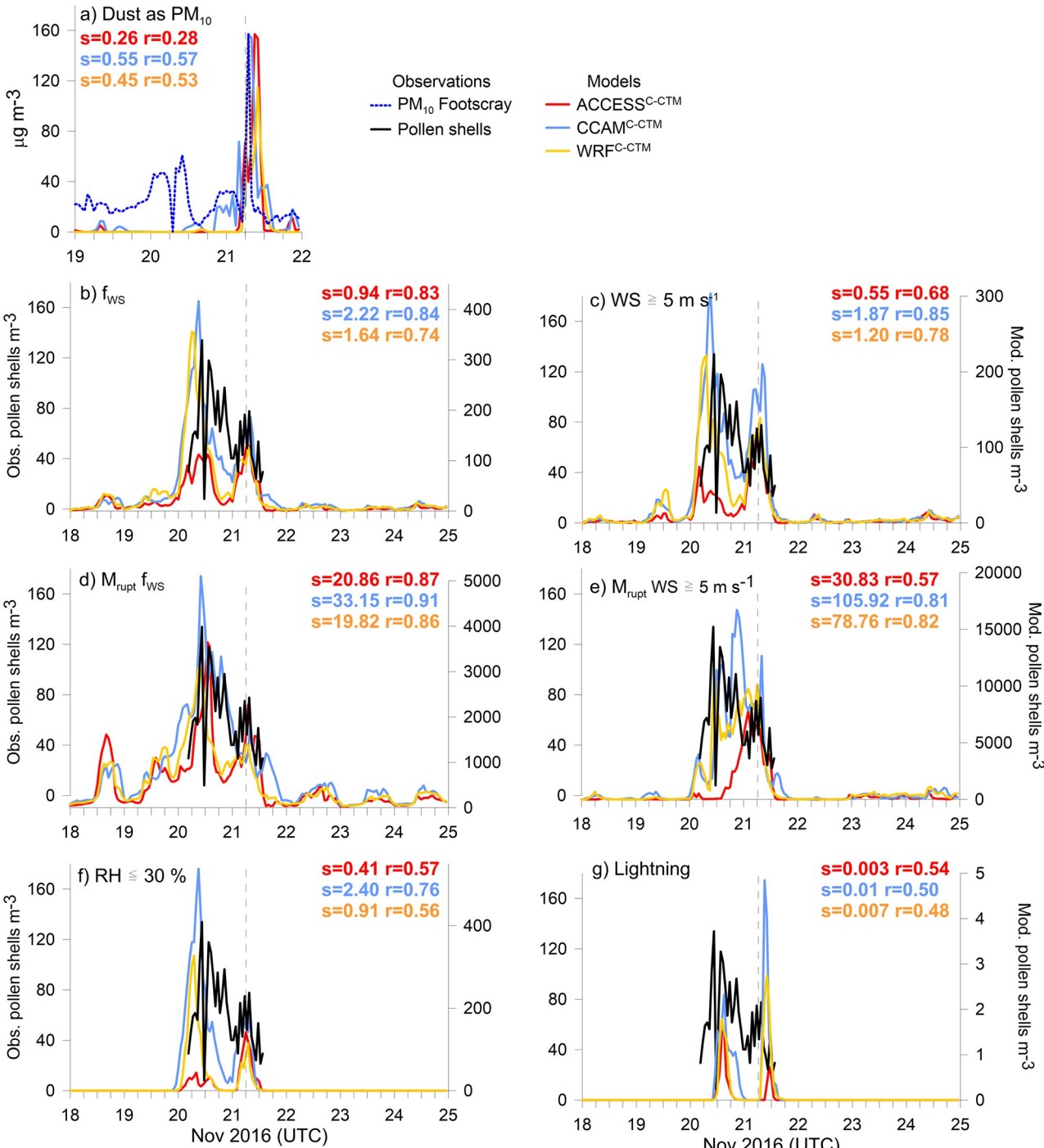

**Fig 6. Time series at Melbourne in (a) dust as PM₁₀, and (b-g) pollen shells for the seven labelled rupturing experiments.** Note the modelled pollen shell concentrations are on the right-hand side y-axes in panels (b-g) and are not similar. s = slope of the linear regression and r = r correlations between modelled and observed pollen shells (or dust), given in same colours as the legend. Vertical dashed line indicates the time of the storm.

currently no model cap to the amount of pollen on the plants. The purpose of the experiment was to investigate whether a rapid increase in pollen shells could be achieved at the correct time. A parameter representing the SPP reservoir can be added when measurements of SPP concentrations are available.

We now review the possibility of pollen rupturing due to electrical activity when the RH is low. In the hours prior to the storm, RH was low in the observations and in all the models, particularly at the surface, and was < 20% in CCAM and WRF up to a height of ~2 km. The timings of the peaks produced by the RH $\leq$ 30% experiment (Fig 6f) are very similar to the in-atmosphere $f_{WS}$ experiment (Fig 6b), but with lower correlations (r = 0.56–0.76 Fig 6f). A peak in pollen shells is produced at the time of the storm, but it is not as large as the peak produced the day before.

Electrical activity due to lightning is now explored as a trigger for rupturing. Fig 7 shows that the pattern of lightning strikes traversed from north west Victoria to south of the city of Melbourne during the day of the storm, and that there were no counts of lightning in the city itself. In this experiment, whole pollen was ruptured to the west of Melbourne, and the pollen shells were transported to the monitoring site (Fig 6g). The models continue to predict two peaks in the time series of pollen shells. CCAM$^{C-CTM}$ and WRF$^{C-CTM}$ produce larger peaks in pollen shells at the time of the storm. However, the predicted concentration of pollen shells in Melbourne is low, partly because the location of the lightning was west of Melbourne, and partly because the parameterisation is dependent on the number of lightning occurrences per grid cell per hour. In this domain the maximum in any hour is only 6.

This paper has focussed on the timeseries of pollen shells measured at Melbourne University. Modelling has shown that lightning or a mechanism based on wind speed provides the best explanation of these measurements. We use Fig 8 to determine the spatial variation in surface SPPs produced at the time of the storm for four of the in-atmosphere rupturing experiments, $f_{WS}$, WS $\geq$ 5 m s$^{-1}$, RH $\leq$ 30% and lightning. Resulting SPP concentrations are in the range $1.4 \times 10^4$–$1.4 \times 10^5$ m$^{-3}$. The aim is to determine if these mechanisms produce a field in SPPs that resembles the position of the storm front, allowing for possible small timing differences related to the meteorology (c.f. Fig 3). There are only SPPs present in the WRF$^{C-CTM}$ atmosphere at this time using the RH $\geq$ 80% threshold option, with the highest concentrations ~$5 \times 10^3$ m$^{-3}$ situated on the coast (S2 Fig in S1 File). The first three experiments ($f_{WS}$, WS $\geq$ 5 m s$^{-1}$ and RH $\leq$ 30%) cause similar SPP fields than the lightning experiment. The first three experiments show SPPs both behind and ahead of the storm front in all three models. The north-south line of the storm is better predicted in ACCESS$^{C-CTM}$, than CCAM$^{C-CTM}$ or WRF$^{C-CTM}$ which show alignment from the north-west to south-east direction. The only experiment to restrict the SPP field behind the storm front occurs using lightning in all three models. At 06:00 UTC the lightning occurred to the west of Melbourne, and the SPPs produced reached Melbourne in the following model timesteps (S3 Fig in S1 File). This is an artefact of the modelling process; in-atmosphere rupturing occurs at the end of each model time step after the airborne concentration of whole pollen is calculated. Transport then occurs at the beginning of the next time step. Following the pattern of lightning occurrences, the position of the peak SPPs incorrectly suggests that towns to the west of Melbourne (e.g. Creswick and Geelong) would receive the majority of the exposure. Although lightning indicates the presence of a thunderstorm, perhaps another meteorological variable of the storm such as diagnosis of strong atmospheric convergence lines might provide a better description of in-atmosphere rupturing [48].

## 4 Conclusions

Within the constraints of the observations (short time-series of pollen shells and without any measurements of SPPs) and the models (e.g. finite temporal and spatial resolution, accuracy of

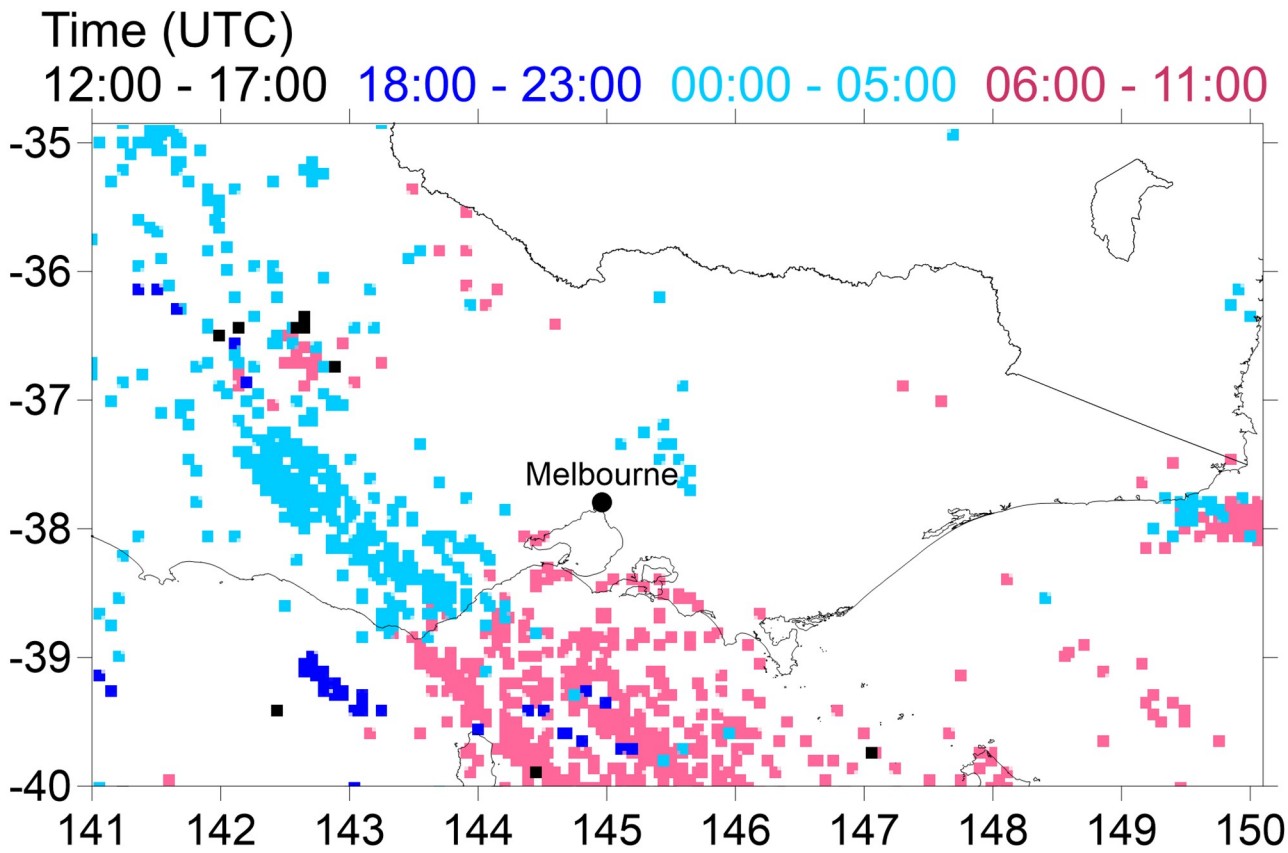

**Fig 7. Location of lightning occurrences in 6-hourly blocks starting late 20 November into 21 November 2016.** The pink pixels coincide with timing of the storm passing Melbourne. Basemap from Igismap.com 2020.

the meteorology and the pollen emission, etc) we have explored possible mechanisms which could have caused the thunderstorm asthma event in Melbourne on 21 November 2016. The commonly cited pollen rupturing mechanism by which whole pollen absorbs water under high RH conditions and is subjected to osmotic shock did not produce peaks in pollen shell concentrations at the time of the storm. The RH was very low prior to the storm, both at the surface ($< 20\%$) and at altitude. Humidity induced rupturing tended to occur at night, and in higher incidences on 19–20 November which did not coincide with reported health impacts.

We do not have measurements of the SPP concentrations in the Melbourne atmosphere at the time of the storm, therefore we cannot ascertain whether the concentrations of pollen shells and SPPs are correlated. Assuming that SPPs are responsible for the severe allergic reaction in humans, we investigated a dust tracer and six other mechanisms occurring in the atmosphere and on the plant, which might cause whole pollen grains to rupture at the time of the storm.

The model simulated a peak in dust particles to occur at the time of the storm, which also coincided with a peak in observed $PM_{10}$ concentrations. This experiment gave rise to several wind-related in-atmosphere and mechanical rupturing experiments. Both in-atmosphere wind experiments caused a larger peak to occur 24 hours before the storm. On-plant mechanical rupturing using a wind speed function, $f_{WS}$ led to the best descriptor of the time series in observed pollen shells, but the peak did not coincide with the storm. On-plant mechanical rupturing using a wind speed threshold of 5 m s$^{-1}$ produced very high concentrations of pollen

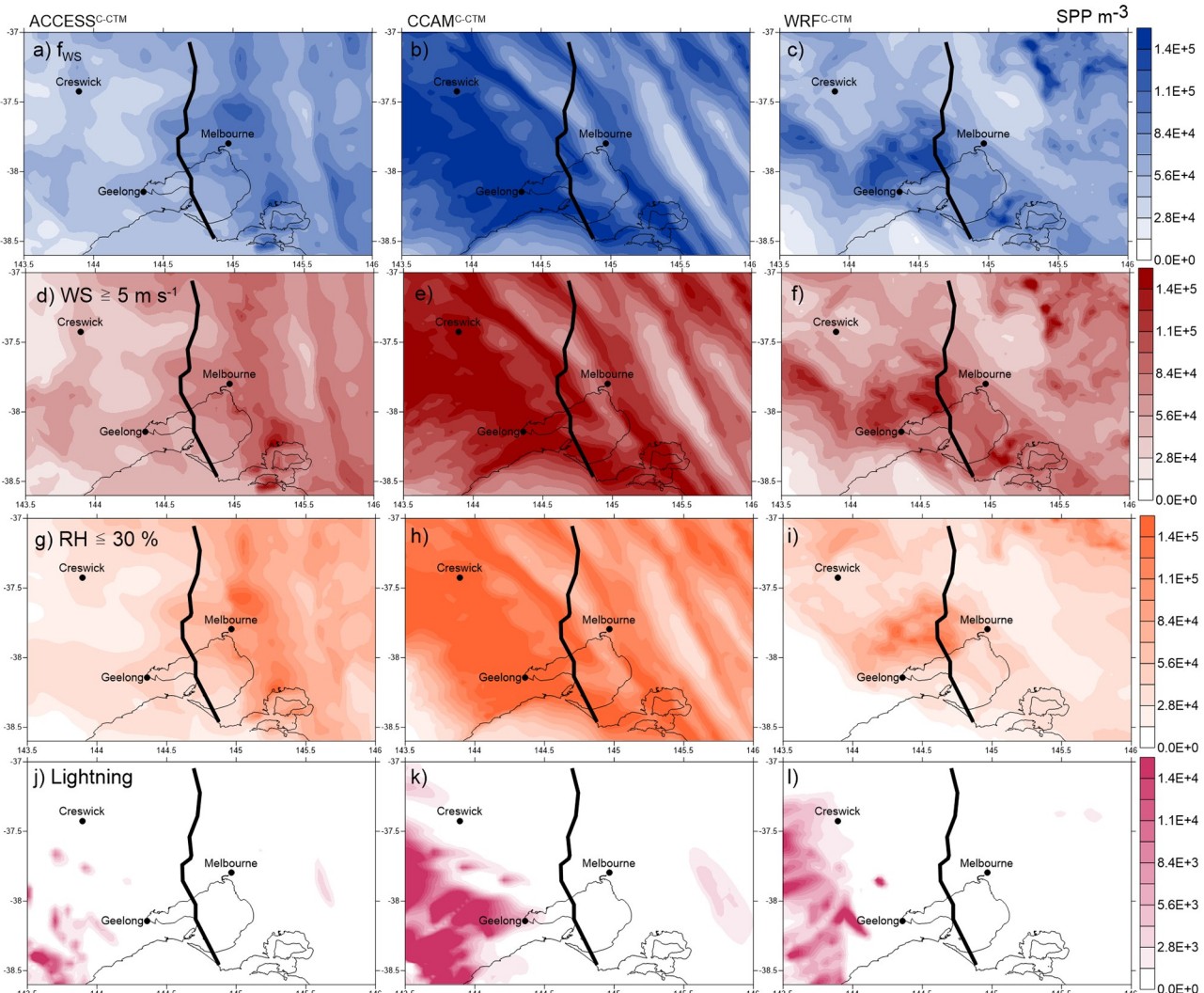

**Fig 8. Sub-pollen particles (m$^{-3}$) in the surface atmosphere at 06:00 UTC.** Does not include mechanical rupturing. The heavy black line shows the approximate position of the storm front diagnosed from radar [19]. Panels (a-c) f$_{WS}$, (d-f) WS > 5 m s$^{-1}$ (g-i) RH < 30% and (j-l) lightning. Note the different scale for lightning colourbar. ACCESS$^{C\text{-}CTM}$ is shown in the left hand panels (a,d,g,j), CCAM$^{C\text{-}CTM}$ in the middle panels (b,e,h,k) and WRF$^{C\text{-}CTM}$ in the right hand panels (c,f,I,l). Basemap from Igismap.com 2020.

shells at the right time. It is plausible that some form of wind action could have contributed to the asthma event, but the relative contributions of ryegrass pollen, fungal spores, wind-blown dust or some other component, alone or in combination, is unknown.

The in-atmosphere rupturing method that produced model peaks in pollen shells at the time of storm were driven by lightning counts. This method also tended to restrict the spatial field in SPPs to behind the storm front, which more accurately describes how the pattern of emergency calls for ambulances evolved after the storm [49].

All the experiments in this study tested aspects of the meteorology that might induce pollen rupturing. However, whilst it is not uncommon for the humidity or wind speed to be high, or there to be lightning associated with a thunderstorm during the peak in the grass pollen season, TA events are uncommon, with an estimated frequency of once every five years even in highly affected cities such as Melbourne. Inclusion of these tested methods into a routine

dynamical pollen forecast system would likely lead to high incidences of false positives in TA risk. A high false positive rate for rare phenomena like thunderstorm asthma is likely to result in 'warning fatigue' from the public. Some new promising research suggests that a thunderstorm itself does not need to be present to cause epidemic asthma, only evidence of strong atmospheric convergence [48]. Very high-resolution modelling experiments to simulate the interaction of pollen with atmospheric boundary layer and in-cloud processes would provide additional insight on the efficacy of the hypothesised mechanism for pollen rupture shown in Fig 1.

The uncertainties highlighted by this work suggest that further targeted laboratory studies of pollen rupture would be of considerable value to help constrain the parameterisation of this process. We also need to understand how many SPPs are required to produce an asthma response and the circumstances in the lifecycle of an SPP which increase their allergenic properties.

## Supporting information

**S1 File.**
(DOCX)

**S2 File.**
(XLSX)

## Acknowledgments

Thanks to Matthew Wozniak of University of Michigan for helpful comments when implementing their pollen rupturing code.

## Author Contributions

**Conceptualization:** Kathryn M. Emmerson.

**Data curation:** Andrew Dowdy, Edward J. Newbigin, Beau W. Picking, Jason Choi.

**Formal analysis:** Kathryn M. Emmerson.

**Funding acquisition:** Elizabeth Ebert.

**Investigation:** Kathryn M. Emmerson.

**Methodology:** Kathryn M. Emmerson, Penelope J. Jones.

**Software:** Jeremy D. Silver, Marcus Thatcher, Alan Wain.

**Writing – original draft:** Kathryn M. Emmerson.

**Writing – review & editing:** Jeremy D. Silver, Penelope J. Jones, Elizabeth Ebert, Tony Bannister.

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
