## [Decision Letter · Decision Letter 0]

28 Oct 2020

PONE-D-20-27816

Atmospheric modelling of grass pollen rupturing mechanisms for thunderstorm asthma prediction.

PLOS ONE

Dear Dr. Emmerson,

Thank you for submitting your manuscript to PLOS ONE. After careful consideration, we feel that it has merit but does not fully meet PLOS ONE’s publication criteria as it currently stands. Therefore, we invite you to submit a revised version of the manuscript that addresses the points raised during the review process.

We look forward to receiving your revised manuscript.

Kind regards,

Chon-Lin Lee, Ph.D.

Academic Editor

PLOS ONE

Journal Requirements:

2.We note that [Figure(s) S1, 7 and 8] in your submission contain [map/satellite] images which may be copyrighted. All PLOS content is published under the Creative Commons Attribution License (CC BY 4.0), which means that the manuscript, images, and Supporting Information files will be freely available online, and any third party is permitted to access, download, copy, distribute, and use these materials in any way, even commercially, with proper attribution. For these reasons, we cannot publish previously copyrighted maps or satellite images created using proprietary data, such as Google software (Google Maps, Street View, and Earth). For more information, see our copyright guidelines: http://journals.plos.org/plosone/s/licenses-and-copyright.

1.    You may seek permission from the original copyright holder of Figure(s) [S1, 7 and 8] to publish the content specifically under the CC BY 4.0 license. 

Reviewers' comments:

Reviewer's Responses to Questions

**Comments to the Author**

1. Is the manuscript technically sound, and do the data support the conclusions?

Reviewer #1: Yes

Reviewer #2: Yes

Reviewer #3: Yes

Reviewer #4: No

2. Has the statistical analysis been performed appropriately and rigorously? 

Reviewer #1: N/A

Reviewer #2: N/A

Reviewer #3: Yes

Reviewer #4: No

3. Have the authors made all data underlying the findings in their manuscript fully available?

Reviewer #1: Yes

Reviewer #2: Yes

Reviewer #3: Yes

Reviewer #4: Yes

4. Is the manuscript presented in an intelligible fashion and written in standard English?

Reviewer #1: Yes

Reviewer #2: Yes

Reviewer #3: Yes

Reviewer #4: Yes

5. Review Comments to the Author

Reviewer #1: PONE-D-20-27816

Atmospheric modelling of grass pollen rupturing mechanisms for thunderstorm asthma prediction.

This paper describes an atmospheric modeling approach to investigate the likely mechanisms of pollen grain breakage under the specific conditions of the November 2016 thunderstorm asthma episode in Melbourne. The work is presented in a clear and understandable way. The article is overall pleasant to read and the conclusions are clearly highlighted.

Although the mechanisms of rupture of pollen grains in the aqueous phase or after exposure to moisture are widely documented in the literature, there is still, in my opinion, a lack of validation in real conditions of these mechanisms of allergen release in the fine fraction of the atmospheric aerosol, especially in stormy conditions. The approach by modeling atmospheric conditions and pollen grain rupture conditions is particularly interesting and brings new information for the aerobiology community. This study is the first to compare the amplitude of the different possible mechanisms of pollen grain rupture in real conditions. In my opinion, this work sheds new light on the question of pollen rupture in thunderstorm conditions, which should provoke a discussion in the 'pollen community', and this, even if some choices of parameterization are probably questionable. I am therefore in favor of a publication of this work but I have a series of questions for which I would like to have a discussion with the authors.

I will first outline these important questions that need to be answered and then in a second part I will list some minor points that could improve the article.

NB: I would like to make it clear that my review of the article does not include parameters related to meteorological models for which I am not a specialist.

Major issues.

• As I understand it, the value of Frupt is based on Taylor's work (2002) and Frupt has been set at 70%. However, it is not clear in the article whether the Frupt value is the same in the equations in lines L155, L178, L211, L221, L243. If the same value of 0.7 was taken for all rupture mechanisms, how can this be justified when it is a percentage breakage of pollen in water after 5 minutes of immersion. Frupt is for example probably different for mechanical or electrical stress.

• L161 – 166. The explanation of the calculation of nspg seems a little confused to me. SPPs include two main types of particles; starch granules and p-particles. The number of 700 to 1000 SPPs corresponds to starch granules (diameter 1.1 µm) (Suphioglu and Abu Chakra). The value of 106 corresponds to p-particles (diameter 0.3-0.4 µm) whose number is more difficult to estimate (Grote 1994, Heslop-Harrison 1982). The diameter value chosen by the authors seems to me too low (200 nm), and if this value is increased to 400 nm diameter, I am afraid that the number of one million p-particles induces a mass of SPPs greater than that of the whole pollen grain. I would therefore like to have the authors' point of view on these considerations and on their choices concerning the size and number of SPPs per pollen grain.

• Fig. 2b and L364-371. By examining the observed shell pollen concentration, it can be seen that it is not maximum at the time of the storm. Is this measurement of shell pollen with a Hirst reliable? Furthermore, if the rupture mechanism does not allow the presence of aerosolized shell pollen, the concentration of the aerosolized shell pollen in the air will not give any information on the concentration of SPPs.

• The mechanism for on-plant mechanical rupturing is not clear to me. How can the pollen rupture on plants because of wind? Can you please better describe this mechanism L218.

Minor issues.

• L151. A reference for the pollen mass is missing.

• Table 1. It would be relevant to add a first column with the description of the rupture mechanism (e.g. 'wet rupture' for RH>80%?). Is the line 'Dust' really useful in this table? Is there a misspelling in line “Mrupt WS” for column “Nrupt” ?

• Fig 1. Flowering grass -> flowering grasses?

• Fig. 5. Add ‘RH’ before %

• Fig 5. Is the unit “log10 SPP grains m-3” means log10 SPPs.m-3” ?

• The discussion on figure 8 should have a more comprehensive conclusion. This part is to my opinion less clear than the rest of the paper.

Reviewer #2: TITLE: Atmospheric modelling of grass pollen rupturing mechanisms for thunderstorm asthma

prediction

The manuscript entitled “Atmospheric modelling of grass pollen rupturing mechanisms for thunderstorm asthma prediction” explored different models to investigate factors influencing pollen rupture that produced high concentrations of SPPs and led to the thunderstorm asthma event in Melbourne. The study used valuable data obtained from several credible sources and demonstrated specialized expertise in this area. Although the data were limited to the locality and the isolated event in 2016, the study helps explain the discrepancies between previous predictions/explanations about pollen rupture mechanisms and the actual phenomenon and will be of interests to scholars worldwide. All data are available without restrictions. Due to the nature of the study, conventional statistical methods were mostly inapplicable.

My main recommendations for improving the manuscript are about making this work more accessible for general readers, especially general practitioners and allergists who will greatly benefit from reading this study. In particular,

The description of the three models in paragraph starting on LINE 123 should be accompanied by a table or a figure. Also, additional information about the similarity/differences between these models should be appreciated.

Several acronyms should be described when first mentioned e.g. C-CTM (LINE96), NCEP FNL analyses (LINE 132).

Overall, there is very little discussion about similarities and differences between the models used in this study compared to other previous models. For example, even though the model from Wozniak et al. was described several times in the methods section, it was never mentioned in the discussion.

LINE 92-93: “In this paper” was repeated twice. Please remove one.

LINE 96; “C-CTM” was introduced for the first time without any explanation. Perhaps, the description in LINE 123-124 should be moved to the Introduction section.

LINE 103: For consistency, “AEDT” should be changed to “UTC”.

LINE 475-487: This paragraph should belong to the discussion section rather than the conclusion section.

In summary, I would recommend a MINOR REVISION for this manuscript.

Reviewer #3: Asthma related to thunderstorms (TA) is one of the phenomena that represents a threat to human health which needs to be deeply studied for a correct prevention. Exacerbations of asthma appearing during a thunderstorm in pollen seasons are characterized, at the beginning of the storms, by a rapid increase in visits to the general practitioners or in the emergency departments of hospitals due to attacks of asthma in subjects with allergic IgE-mediated sensitization to allergenic pollens (prevalently Gramineae and Parietaria) or mold spores of Alternaria alternata. In the first 20 to 30 minutes of a thunderstorm, patients with pollen allergy can inhale a high concentration of allergens released in atmosphere from pollen grains after the rupture of pollens. Patients without asthma symptoms, but who suffer from seasonal rhinitis may have an asthma attack during a thunderstorm in pollen seasons .

During the events of TA there is a strong association with the elevation of atmospheric concentrations of pollen grains, such as grasses or other species of allergenic plants and symptoms of respiratory allergy and asthma attacks. . A possible explanation for TA involves the role of rainwater promoting the release of inhalable particles deriving from the rupture of pollen grains. The world’s most severe thunderstorm asthma event occurred in Melbourne, Australia on 21 November 2016, coinciding with the peak of the grass pollen season with 10 deaths and about one twousand persons with asthma attacks.

As mentioned before, the aetiological role of thunderstorms in these events is thought to derive from the rupture of pollens in high humidity conditions, releasing large numbers of sub-pollen particles (SPPs) with pauci-micronic size very easily inhaled deep into the lungs. In this manuscript the authors try to explain the pathogenetic background of TA. The humidity hypothesis was implemented into a three dimensional atmospheric model and driven by inputs from three meteorological models, but the mechanism could not explain how the Melbourne event occurred because relative humidity was very low throughout the atmosphere. Tests in this paper showed humidity induced rupturing occurred frequently at other times and would likely lead to recurrent false alarms if used in a predictive capacity. They used the model to investigate a range of other possible pollen rupturing mechanisms which could have produced high concentrations of subpollen particles in the atmosphere during the storm. The mechanisms studied involve mechanical friction from wind gusts, electrical build up and discharge incurred during conditions of low relative humidity, and lightning strikes.

In particular the mechanisms studied involve mechanical friction from wind gusts, the humidity and the electrical role of thunderbolds ( lightning strikes) . The results of this study suggest that these mechanisms likely operated in tandem with one another, but the lightning method was the only mechanism to generate a pattern in SPPs following the path of the storm. If humidity induced rupturing cannot explain the 2016 Melbourne event, then new targeted laboratory studies of alternative pollen rupture mechanisms would be of value to help constrain the parameterisation of the pollen rupturing process.

In my opinin this is an intersting update on pathogenetic mechanisms of TA but

I suggest to add considerations on the clinical importance of these events to better understand the pathogenesis of these event to prevent in future the risk of near fatal asthma and of deaths for asthma.

Considering that these events occurred not only in Australia but exacerbations and asthma epidemics related to thunderstorms have been described in several cities, mainly in Europe (Birmingham and London in the United Kingdom and Naples in Italy), I suggest to add in the references also manuscripts published in journals of American Academy AAAAI such as JACI ( D'Amato G, et al Thunderstorm-related asthma attacks .J Allergy Clin Immunol. 2017 Jun;139(6):1786-1787. doi: 10.1016/j.jaci.2017.03.003. Epub 2017 Mar 23.PMID: 28342913 ) and European Academy EAACI ( D'Amato G, et al . Latest news on relationship between thunderstorms and respiratory allergy, severe asthma, and deaths for asthma. Allergy. 2019;74(1):9-11.)

-

I do not suggest to accept the paper considering that authors haven’t modified the manuscript following my suggestions. You asked my engagemend as reviewer, I worked a lot to read the paper and to suggest some variations and integrations that authors haven’t accepted my suggestions, in particular, they havent’ accepted to integrate references with two recent publications publishd in Journals of American Academy of Allergy, Asthma and Clinical Immunology (JACI) and in the journal of European Academy (Allergy). In my opinion, as invited reviewer , the paper can’t be accepted for publication without this integration of citations.

Reviewer #4: This manuscript uses atmospheric models to test and predict the concentrations of pollen grains and sub-pollen particles (SPP) in the atmosphere surrounding the 20-21 November thunderstorm asthma epidemic in Melbourne, Australia. The article addresses an important topic and concludes that a better understanding of pollen rupturing under laboratory conditions is needed to develop accurate predictive models.

Several aspects of the manuscript are in need of significant improvement prior to consideration for publication. In particular, the observational data should be expanded well beyond the 34 hours surrounding the thunderstorm asthma epidemic to enable more robust model evaluation at times when asthma was not triggered, but pollen counts may have been high. Further, more robust quantitative analysis of model performance in reproducing observations is needed. Model assumptions, particularly the number of SPP per pollen grain needs further consideration and justification. Some highly relevant and recently literature should be considered.

1. The use of only 34 hours of intact pollen grain counts and pollen shell data is much too limited. For the results presented herein to be considered robust, the authors must expand the time period of experimental observations to include additional time periods before and after this period. One week should be considered the minimum duration for consideration. This is particularly important to evaluation of the model accuracy. Further, it is critical to demonstrating how the model could over-predict SPP outside of the selected thunderstorm asthma event.

2. In exploring different scenarios that could contribute SPP to the atmosphere (section 3.2 and figure 6), it is critical for the authors to consider time periods outside of the 34 hours to assess model accuracy. For example, do the models suggest wind speeds > 5 m/s outside of the 21 November event contribute to pollen shells?

3. Throughout the manuscript the authors refer to correlations between the model and observations (e.g., line 368, 378, 380, 383, 403, 430, etc.), but no correlation coefficients are presented. Quantitative comparisons to evaluate the model accuracy are needed.

4. The approximation by Wozniak et al. 2018 of 106 SPP per pollen grain was intended to be an upper estimate of the number of cloud condensation nuclei (CCN) to estimate the maximum potential impact of SPP on precipitation. It is not a realistic estimate, especially in light of the SPP size distributions observed experimentally for ryegrass, which includes a large number of starch granules 1-2 μm in diameter (Suphioglu et al. 1992, Taylor et al. 2002) that dominate the mass of SPP. Consequently, the chosen nSPP is a large over-estimate of the number of SPP per ryegrass pollen grain. Further discussion of experimental data and justification for the assumptions made are needed.

5. It is incorrect that SPP cannot be monitored with current techniques (lines 279 and 483). Hughes et al. (2020) report the first online measurements of SPP (a.k.a. pollen fragments) during convective storms using single-particle fluorescence spectroscopy and chemical tracers.

6. The findings of Hughes et al. (2020) should be considered and discussed in the context of the model evaluation. Key points to consider is the timing of SPP concentrations with respect to the arrival of the thunderstorm (e.g., line 289), their observation of SPP in the absence of lightning, concentrations of SPP observed relative to the model (e.g., line 447), their observations of higher SPP concentrations in strong storms with high windspeeds and downdrafts (which may help to explain differences across 20 and 21 November (e.g., line 479), and their observations of SPP in precipitation events of many types during tree pollen season (which suggests SPP events are quite common, but vary in strength; line 505-507).

7. Interestingly, Hughes et al. (2020) estimate that at least one-third of the pollen grains ruptured in the convective storm highlighted in their article, similar to the fraction of total pollen grains that the pollen shells in this study, suggesting similarities in the number of pollens that rupture in the case of ryegrass in Melbourne.

8. The model described appears to allow for pollen rupturing at high humidity (section 3.1). Does the model consider and account for the closing of anthers at high humidity to minimize pollen exposure to humid air? This is a likely source of error contributing to the high estimates of SPP at night that is not consistent with observations.

9. In stating “we find that the RH throughout the atmosphere to be well below the 80% required for pollen altitude described in Taylor and Jonsson (2004), which is suggested to approach 100% at the cloud base? For example, in Figure 4, it appears that the 80% threshold is met by some models at a pressure of 600 hPa, which is relevant to pollen grains entrained in updrafts. The statement in the abstract that most pollen remained within 40 m of the surface at line 24 suggests that the model may not accurately represent the vertical distribution of pollen grains in the atmosphere.

10. Like 396 – the statement that SPP are too small to contribute much to PM2.5 is not supported with any evidence. Meanwhile, prior work by Rathnayake et al. (2017) showed that pollen contributed 0.74 μg/m3 corresponding to 42% of PM2.5 mass on 2 May 2013 during tree pollen season when a thunderstorm struck.

11. In considering the role of lightning (paragraph beginning at line 433) – have the authors considered cloud-to-cloud lightning, or only cloud-to-ground lightning strikes? Both may be relevant to pollen rupturing, particularly to pollen grains at higher altitudes.

12. Line 49 – Provide an explanation for how SPP would become concentrated at ground level.

13. Figure 1 – Define the meaning of the symbols used, especially different types of arrows, plusses, and minuses.

14. Table 1 – It would be helpful to the reader if you defined the many variables in these equations as part of the table.

15. Terminology – pollen “exine” is a more biologically accurate term than “shell”

Works Cited

Hughes, D. D., C. B. A. Mampage, L. M. Jones, Z. Liu and E. A. Stone (2020). "Characterization of Atmospheric Pollen Fragments during Springtime Thunderstorms." Environmental Science & Technology Letters 7(6): 409-414.

Rathnayake, C. M., N. Metwali, T. Jayarathne, J. Kettler, Y. Huang, P. S. Thorne, P. T. O'Shaughnessy and E. A. Stone (2017). "Influence of rain on the abundance of bioaerosols in fine and coarse particles." Atmos. Chem. Phys. 17(3): 2459-2475.

Suphioglu, C., M. B. Singh, P. Taylor, R. Bellomo, P. Holmes, R. Puy and R. B. Knox (1992). "Mechanism of grass-pollen induced asthma." Lancet 339(8793): 569-572.

Taylor, P. E., R. C. Flagan, R. Valenta and M. M. Glovsky (2002). "Release of allergens as respirable aerosols: A link between grass pollen and asthma." Journal of Allergy and Clinical Immunology 109(1): 51-56.

Taylor, P. E. and H. Jonsson (2004). "Thunderstorm asthma." Current Allergy and Asthma Reports 4(5): 409-413.

6. PLOS authors have the option to publish the peer review history of their article (what does this mean?). If published, this will include your full peer review and any attached files.

Reviewer #1: **Yes: **Nicolas Visez

Reviewer #2: No

Reviewer #3: **Yes: **Prof Gennaro D'Amato

Reviewer #4: No

---

## [Author Response · Author response to Decision Letter 0]

7 Dec 2020

We thank the editor and reviewers for their consideration of our manuscript and use their comments to make our revised manuscript clearer. Our responses to their comments are in blue text.

Editor comments

Done

3.We note that [Figure(s) S1, 7 and 8] in your submission contain [map/satellite] images which may be copyrighted. 

The Australian coast and State border printed in the maps in figures S1, 7 and 8 were produced using free shape files from https://www.igismap.com/australia-shapefile-download/ and are not copyrighted (Open Data Commons Open Database License (ODbL)). I marked the towns of Geelong, Creswick and Melbourne on these maps manually by providing the longitude and latitudes. I will include “Basemap from Igismap.com 2020” in the figure captions.

1. You may seek permission from the original copyright holder of Figure(s) [S1, 7 and 8] to publish the content specifically under the CC BY 4.0 license. 

Figures uploaded and checked by PACE. Downloaded the adjusted figures.

Reviewer #1: PONE-D-20-27816

Atmospheric modelling of grass pollen rupturing mechanisms for thunderstorm asthma prediction.

This paper describes an atmospheric modeling approach to investigate the likely mechanisms of pollen grain breakage under the specific conditions of the November 2016 thunderstorm asthma episode in Melbourne. The work is presented in a clear and understandable way. The article is overall pleasant to read and the conclusions are clearly highlighted.

Although the mechanisms of rupture of pollen grains in the aqueous phase or after exposure to moisture are widely documented in the literature, there is still, in my opinion, a lack of validation in real conditions of these mechanisms of allergen release in the fine fraction of the atmospheric aerosol, especially in stormy conditions. The approach by modeling atmospheric conditions and pollen grain rupture conditions is particularly interesting and brings new information for the aerobiology community. This study is the first to compare the amplitude of the different possible mechanisms of pollen grain rupture in real conditions. In my opinion, this work sheds new light on the question of pollen rupture in thunderstorm conditions, which should provoke a discussion in the 'pollen community', and this, even if some choices of parameterization are probably questionable. I am therefore in favor of a publication of this work but I have a series of questions for which I would like to have a discussion with the authors.

I will first outline these important questions that need to be answered and then in a second part I will list some minor points that could improve the article.

NB: I would like to make it clear that my review of the article does not include parameters related to meteorological models for which I am not a specialist.

Major issues.

1) As I understand it, the value of Frupt is based on Taylor's work (2002) and Frupt has been set at 70%. However, it is not clear in the article whether the Frupt value is the same in the equations in lines L155, L178, L211, L221, L243. If the same value of 0.7 was taken for all rupture mechanisms, how can this be justified when it is a percentage breakage of pollen in water after 5 minutes of immersion. Frupt is for example probably different for mechanical or electrical stress.

The value of Frupt is 0.7 for all experiments, as there is no literature on the rupturing rate of ryegrass pollen for other mechanisms. By keeping the rupturing rate the same across all simulations, it allows us to directly compare the amplitudes of the different mechanisms. I realise that we didn’t actually specify the 0.7 in the text. 

At line 156 ‘where Frupt is the fraction of whole grass pollen grains that rupture (=0.7). Frupt is the same for all simulations in the absence of literature on other rates of ryegrass pollen rupturing.’

2) L161 – 166. The explanation of the calculation of nspg seems a little confused to me. SPPs include two main types of particles; starch granules and p-particles. The number of 700 to 1000 SPPs corresponds to starch granules (diameter 1.1 µm) (Suphioglu and Abu Chakra). The value of 106 corresponds to p-particles (diameter 0.3-0.4 µm) whose number is more difficult to estimate (Grote 1994, Heslop-Harrison 1982). The diameter value chosen by the authors seems to me too low (200 nm), and if this value is increased to 400 nm diameter, I am afraid that the number of one million p-particles induces a mass of SPPs greater than that of the whole pollen grain. I would therefore like to have the authors' point of view on these considerations and on their choices concerning the size and number of SPPs per pollen grain.

The values used were those cited in the Wozniak et al. 2018 paper. However the model is very quick to re-run. Similar to comment 19, we have decided to change the parameters to be specific to the ryegrass measurements of Suphioglu et al (1992), and are now using nspg = 700, diameter of SPP = 600 nm which is the lower end of Suphioglu et al (1992) measurements (600 nm – 2.5 �m) to be conservative. Use of the 600 nm diameter means the mass of 1 SPP is now 1.13 x 10-13 g. 

Please note that most of the figures in the paper use the number of whole pollen ruptured, Nrupt which does not change with these new SPP parameters. 

We alter the text starting at line 158 to say ‘ The measurements of Suphioglu et al. (1) on ryegrass showed around 700 SPPs per whole pollen grain of a size range 600 nm – 2.5 �m. We will use 700 SPPs per whole pollen grain with an SPP diameter of 600 nm which is the lower end of Suphioglu et al’s (16) measurements. The SPPs have the same density as whole grass pollen. This yields a mass of one SPP, mSPP = 1.13 × 10-13 g and a fall speed of 0.01 cm s-1.’

3) Fig. 2b and L364-371. By examining the observed shell pollen concentration, it can be seen that it is not maximum at the time of the storm. Is this measurement of shell pollen with a Hirst reliable? Furthermore, if the rupture mechanism does not allow the presence of aerosolized shell pollen, the concentration of the aerosolized shell pollen in the air will not give any information on the concentration of SPPs.

The Hirst or Burkard spore trap remains the standard way to measure pollen. At line 102 “using standard methods”

Other researchers, both published and unpublished have observed asthma events occurring without a correlation with broken pollen shells (see line 366).

We state in section 2.1 that we only have observations of the shell pollen and not the SPPs. We do not know how many SPPs were in the air at the time of the storm. What the paper is trying to do is create a peak in Nrupt at the time of the storm and then hypothesise the concentration of SPPs from this.

4) The mechanism for on-plant mechanical rupturing is not clear to me. How can the pollen rupture on plants because of wind? Can you please better describe this mechanism L218.

Wozniak et al (2018) describe a mechanism for pollen rupturing on the plant using the precipitation and relative humidity functions. What we have done is replace these parameters with wind speed to describe the process of sweeping particles into the atmosphere. The key point is that Mrupt is not reliant on pollen already being airborne in order to rupture it, as it is an emission from the ground. This could account for resuspension of deposited SPPs.

We add text at line 222: ‘A prerequisite of in-atmosphere rupture is the presence of airborne pollen that is already disperse. We adjust Wozniak et al’s description of mechanical rupturing from Eq. 2 to represent a wind speed induced emission of 600 nm particles from the ground, replacing (1-fPRfRH) with either fWS or wind speed as the part of the definition, e.g.:

M_rupt=〖F_rupt n_spg P〗_fx f_WS m_SPP/m_pol , (8) 

This process considers the resuspension of previously deposited SPPs.’

Minor issues.

5) L151. A reference for the pollen mass is missing.

At line 151 ‘Derived in Emmerson et al. (2019)’

6) Table 1. It would be relevant to add a first column with the description of the rupture mechanism (e.g. 'wet rupture' for RH>80%?). Is the line 'Dust' really useful in this table? Is there a misspelling in line “Mrupt WS” for column “Nrupt” ?

We’d like to keep the dust line in the table as it is a model simulation. We have replaced the threshold column with a description of the simulation.

Yes, there was a misspelling, it is corrected now.

Experiment name Description In-atmosphere rupture, Nrupt On-plant mechanical rupture, Mrupt

RH ≥ 80% Humidity induced rupturing when RH is ≥ 80% F_rupt×χ 

Mrupt RH ≥ 80% F_rupt× χ 〖F_rupt n_spg P〗_fx (1-f_PR f_RH ) m_SPP/m_pol 

fWS Rupturing (or emission) using a function of wind speed F_rupt× χ 〖×f〗_WS 

Mrupt fWS F_rupt× χ× f_WS 〖F_rupt n_spg P〗_fx f_WS m_SPP/m_pol 

WS ≥ 5 m s-1 Rupturing (or emission) using wind speeds ≥ 5 m s-1 F_rupt× χ×WS 

Mrupt WS F_rupt ×χ ×WS 〖F_rupt n_spg P〗_fx WS m_SPP/m_pol 

RH ≤ 30% Static electricity induced rupturing when RH is ≤ 30% F_rupt× χ 

Lightning Lightning induced rupturing F_rupt×χ〖 ×n〗_lightning 

Dust Uses a dust tracer Not applicable 

7) Fig 1. Flowering grass -> flowering grasses?

Done

8) Fig. 5. Add ‘RH’ before %

Done

9) Fig 5. Is the unit “log10 SPP grains m-3” means log10 SPPs.m-3” ?

Corrected to log10 SPP m-3

10) The discussion on figure 8 should have a more comprehensive conclusion. This part is to my opinion less clear than the rest of the paper.

Line 445 ‘This paper has focussed on the timeseries of pollen shells measured at Melbourne University. Modelling has shown that lightning or a mechanism based on wind speed provides the best explanation of these point measurements. We use figure 8 to determine the spatial variation in SPPs produced at the time of the storm from each of the model simulations, with the aim being to determine if the mechanisms produce a field in SPPs that resembles the position of the storm front, allowing for possible small timing differences related to the meteorology (c.f. Figure 3). ‘

Line 458 ‘At 06:00 UTC the lightning occurred to the west of Melbourne, and the SPPs produced reached Melbourne in the following model timesteps (supplementary figure S2). This is an artefact of the modelling process; in-atmosphere rupturing occurs at the end of each model time step after the airborne concentration of whole pollen is calculated. Transport then occurs at the beginning of the next time step. Following the pattern of lightning occurrences, the position of the peak SPPs incorrectly suggests that towns to the west of Melbourne (e.g. Creswick and Geelong) would receive the majority of the exposure. Although lightning indicates the presence of a thunderstorm, perhaps another meteorological variable of the storm such as diagnosis of strong atmospheric convergence lines might provide a better description of in-atmosphere rupturing (Bannister et al 2020). ’ 

Reviewer #2: TITLE: Atmospheric modelling of grass pollen rupturing mechanisms for thunderstorm asthma

prediction

The manuscript entitled “Atmospheric modelling of grass pollen rupturing mechanisms for thunderstorm asthma prediction” explored different models to investigate factors influencing pollen rupture that produced high concentrations of SPPs and led to the thunderstorm asthma event in Melbourne. The study used valuable data obtained from several credible sources and demonstrated specialized expertise in this area. Although the data were limited to the locality and the isolated event in 2016, the study helps explain the discrepancies between previous predictions/explanations about pollen rupture mechanisms and the actual phenomenon and will be of interests to scholars worldwide. All data are available without restrictions. Due to the nature of the study, conventional statistical methods were mostly inapplicable.

My main recommendations for improving the manuscript are about making this work more accessible for general readers, especially general practitioners and allergists who will greatly benefit from reading this study. In particular,

11) The description of the three models in paragraph starting on LINE 123 should be accompanied by a table or a figure. Also, additional information about the similarity/differences between these models should be appreciated.

Several acronyms should be described when first mentioned e.g. C-CTM (LINE96), NCEP FNL analyses (LINE 132).

Overall, there is very little discussion about similarities and differences between the models used in this study compared to other previous models. For example, even though the model from Wozniak et al. was described several times in the methods section, it was never mentioned in the discussion.

The Wozniak model was built to represent pollen rupture in terms of providing CCN for precipitation, not for thunderstorm asthma purposes. We use their rupturing mechanisms which are the model simulations referred to as RH >80% and Mrupt RH > 80%.

We include this table for the supplementary section. At line 132 ‘Details of meteorological models used are given in supplementary Table S1’.

 ACCESS CCAM WRF

Boundary conditions Global ACCESS ERA Interim NCEP/FNL

Topography Geoscience Australia digital elevation Mapping Geoscience Australia digital elevation Mapping Geoscience Australia digital elevation Mapping

Sea surface temperatures Global Australian Multi-Sensor SST Analysis Daily 0.25° (2)

ERA Interim Real-Time Global NCEP

Boundary layer scheme Lock et al (3) , Edwards et al. (4)

 Prognostic turbulence kinetic energy and eddy dissipation (5)

Mellor-Yamada-Janjic scheme (6)

Microphysics scheme Wilson and Ballard (7) single-moment bulk microphysics scheme; rain as a prognostic variable

 Prognostic condensate scheme (8,9)

Morrison double-moment scheme (10)

Radiation Edwards and Slingo (11) with incremental time stepping scheme (12) 

Longwave: Schwarzkopf and Ramaswamy (13) Shortwave: Freidenreich and Ramaswamy (14)

Rapid radiative transfer model (RRTM) (15)

Land surface scheme The Met Office Surface Exchange Scheme version 2 (MOSES2; Essery et al. (16) 

Kowalczyk et al (17)

Noah (18)

Convection Gregory and Rowntree (19)

Mass-flux closure Grell 3D ensemble Scheme (20)

Aerosol feedbacks No Prognostic aerosols with direct and indirect effects (21)

No

Cloud feedbacks yes Wilson et al. (22)

 yes yes

At line 139 ‘CCAM and WRF meteorological capabilities were evaluated across south east Australia and found to reproduce temperatures and local scale meteorological features well, with wind speeds and water vapour mixing ratios within benchmark ranges (Monk et al. 2019).’

12) LINE 92-93: “In this paper” was repeated twice. Please remove one.

Done

13) LINE 96; “C-CTM” was introduced for the first time without any explanation. Perhaps, the description in LINE 123-124 should be moved to the Introduction section.

We have removed ‘C-CTM’ at line 96, and keep the description on line 123..

NCEP FNL is National Centers for Environmental Prediction Final reanalyses.

14) LINE 103: For consistency, “AEDT” should be changed to “UTC”.

Done

15) LINE 475-487: This paragraph should belong to the discussion section rather than the conclusion section.

Done

In summary, I would recommend a MINOR REVISION for this manuscript.

Reviewer #3: 

Asthma related to thunderstorms (TA) is one of the phenomena that represents a threat to human health which needs to be deeply studied for a correct prevention. Exacerbations of asthma appearing during a thunderstorm in pollen seasons are characterized, at the beginning of the storms, by a rapid increase in visits to the general practitioners or in the emergency departments of hospitals due to attacks of asthma in subjects with allergic IgE-mediated sensitization to allergenic pollens (prevalently Gramineae and Parietaria) or mold spores of Alternaria alternata. In the first 20 to 30 minutes of a thunderstorm, patients with pollen allergy can inhale a high concentration of allergens released in atmosphere from pollen grains after the rupture of pollens. Patients without asthma symptoms, but who suffer from seasonal rhinitis may have an asthma attack during a thunderstorm in pollen seasons .

During the events of TA there is a strong association with the elevation of atmospheric concentrations of pollen grains, such as grasses or other species of allergenic plants and symptoms of respiratory allergy and asthma attacks. . A possible explanation for TA involves the role of rainwater promoting the release of inhalable particles deriving from the rupture of pollen grains. The world’s most severe thunderstorm asthma event occurred in Melbourne, Australia on 21 November 2016, coinciding with the peak of the grass pollen season with 10 deaths and about one twousand persons with asthma attacks.

As mentioned before, the aetiological role of thunderstorms in these events is thought to derive from the rupture of pollens in high humidity conditions, releasing large numbers of sub-pollen particles (SPPs) with pauci-micronic size very easily inhaled deep into the lungs. In this manuscript the authors try to explain the pathogenetic background of TA. The humidity hypothesis was implemented into a three dimensional atmospheric model and driven by inputs from three meteorological models, but the mechanism could not explain how the Melbourne event occurred because relative humidity was very low throughout the atmosphere. Tests in this paper showed humidity induced rupturing occurred frequently at other times and would likely lead to recurrent false alarms if used in a predictive capacity. They used the model to investigate a range of other possible pollen rupturing mechanisms which could have produced high concentrations of subpollen particles in the atmosphere during the storm. The mechanisms studied involve mechanical friction from wind gusts, electrical build up and discharge incurred during conditions of low relative humidity, and lightning strikes.

In particular the mechanisms studied involve mechanical friction from wind gusts, the humidity and the electrical role of thunderbolds ( lightning strikes) . The results of this study suggest that these mechanisms likely operated in tandem with one another, but the lightning method was the only mechanism to generate a pattern in SPPs following the path of the storm. If humidity induced rupturing cannot explain the 2016 Melbourne event, then new targeted laboratory studies of alternative pollen rupture mechanisms would be of value to help constrain the parameterisation of the pollen rupturing process.

In my opinin this is an intersting update on pathogenetic mechanisms of TA but

I suggest to add considerations on the clinical importance of these events to better understand the pathogenesis of these event to prevent in future the risk of near fatal asthma and of deaths for asthma.

Considering that these events occurred not only in Australia but exacerbations and asthma epidemics related to thunderstorms have been described in several cities, mainly in Europe (Birmingham and London in the United Kingdom and Naples in Italy), I suggest to add in the references also manuscripts published in journals of American Academy AAAAI such as JACI ( D'Amato G, et al Thunderstorm-related asthma attacks .J Allergy Clin Immunol. 2017 Jun;139(6):1786-1787. doi: 10.1016/j.jaci.2017.03.003. Epub 2017 Mar 23.PMID: 28342913 ) and European Academy EAACI ( D'Amato G, et al . Latest news on relationship between thunderstorms and respiratory allergy, severe asthma, and deaths for asthma. Allergy. 2019;74(1):9-11.)

I do not suggest to accept the paper considering that authors haven’t modified the manuscript following my suggestions. You asked my engagemend as reviewer, I worked a lot to read the paper and to suggest some variations and integrations that authors haven’t accepted my suggestions, in particular, they havent’ accepted to integrate references with two recent publications publishd in Journals of American Academy of Allergy, Asthma and Clinical Immunology (JACI) and in the journal of European Academy (Allergy). In my opinion, as invited reviewer , the paper can’t be accepted for publication without this integration of citations.

We are confused by this review. Firstly our manuscript has never been submitted or reviewed anywhere else, and the reviewer has mistaken it for another piece of work they have reviewed. As stated in the covering letter, our manuscript has an atmospheric modelling focus, not on the clinical aspects of asthma. We don’t think the manuscript warrants lengthening with such a clinical discussion. 

Reviewer #3 does not have any comments on the science of our manuscript, only that it doesn’t include two of his references. If the editor decides this is an impediment to our publishing this manuscript, then we will include them.

Reviewer #4: 

This manuscript uses atmospheric models to test and predict the concentrations of pollen grains and sub-pollen particles (SPP) in the atmosphere surrounding the 20-21 November thunderstorm asthma epidemic in Melbourne, Australia. The article addresses an important topic and concludes that a better understanding of pollen rupturing under laboratory conditions is needed to develop accurate predictive models.

Several aspects of the manuscript are in need of significant improvement prior to consideration for publication. In particular, the observational data should be expanded well beyond the 34 hours surrounding the thunderstorm asthma epidemic to enable more robust model evaluation at times when asthma was not triggered, but pollen counts may have been high. Further, more robust quantitative analysis of model performance in reproducing observations is needed. Model assumptions, particularly the number of SPP per pollen grain needs further consideration and justification. Some highly relevant and recently literature should be considered.

16. The use of only 34 hours of intact pollen grain counts and pollen shell data is much too limited. For the results presented herein to be considered robust, the authors must expand the time period of experimental observations to include additional time periods before and after this period. One week should be considered the minimum duration for consideration. This is particularly important to evaluation of the model accuracy. Further, it is critical to demonstrating how the model could over-predict SPP outside of the selected thunderstorm asthma event.

We now expand the time series plots in figures 2 and 6 to include 1 full week of model results. 

It is not possible to expand the hourly pollen grain nor pollen shell data, as 24-hour concentrations of pollen measured on days prior to, and after Nov 21st were mostly too low to make hourly counting worthwhile (minimum 50 grain m-3). 

Date 24-hour whole pollen concentration grains m-3

18.11.16 60

19.11.16 22

20.11.16 29

21.11.16 102

22.11.16 19

23.11.16 3

24.11.16 17

We have included these 24-hourly observations in the time series plot of figure 2a.

At line 111. “Usual practice is to count pollen across 24-hour transects, as the process is manually demanding; for this study we counted vertical transects to obtain hourly measurements around the 21 November event, with pollen counts converted into grains m-3. 24 hour pollen counts either side of 21 November were too low to consider hourly subdivision.”

Line 290 ‘The models show that humidity induced rupturing was strongest at other times in the week, e.g. 18 and 19 November (Figure 2b),’

17. In exploring different scenarios that could contribute SPP to the atmosphere (section 3.2 and figure 6), it is critical for the authors to consider time periods outside of the 34 hours to assess model accuracy. For example, do the models suggest wind speeds > 5 m/s outside of the 21 November event contribute to pollen shells?

We have expanded the model timeseries in figure 6 to include 1 week from 18-24 November 2016 (with the exception of the dust simulation). The wind gusts were strongest on 20 - 21 November which is already discussed in section 3.2.

18. Throughout the manuscript the authors refer to correlations between the model and observations (e.g., line 368, 378, 380, 383, 403, 430, etc.), but no correlation coefficients are presented. Quantitative comparisons to evaluate the model accuracy are needed.

Correlation coefficients are presented within the legends of each panel within figures 2 and 6. We include them within the text to make it clearer.

Line 368 (r =0.08 – 0.30) Line 378 (r=0.33 – 0.51 in figure 6d) Line 380 (r=0.03 – 0.30 in figure 6g) Line 383 ‘r correlations’ Line 403 (r=0.33 – 0.51 in figure 6d) Line 430 (r=0.04 – 0.28 figure 6f) 

19. The approximation by Wozniak et al. 2018 of 106 SPP per pollen grain was intended to be an upper estimate of the number of cloud condensation nuclei (CCN) to estimate the maximum potential impact of SPP on precipitation. It is not a realistic estimate, especially in light of the SPP size distributions observed experimentally for ryegrass, which includes a large number of starch granules 1-2 μm in diameter (Suphioglu et al. 1992, Taylor et al. 2002) that dominate the mass of SPP. Consequently, the chosen nSPP is a large over-estimate of the number of SPP per ryegrass pollen grain. Further discussion of experimental data and justification for the assumptions made are needed.

As with comment 2, we have re-run the model using the diameter of 1 SPP = 600 nm and 700 SPP per whole pollen ruptured. These new parameters now represent ryegrass pollen observations. Text changed as per comment 2.

20. It is incorrect that SPP cannot be monitored with current techniques (lines 279 and 483). Hughes et al. (2020) report the first online measurements of SPP (a.k.a. pollen fragments) during convective storms using single-particle fluorescence spectroscopy and chemical tracers.

At line 279 ‘We have not measured SPP concentrations in this work’ 

At line 483 ‘As we cannot determine the concentrations of SPPs in the Melbourne atmosphere at the time of the storm, we do not know whether the concentrations of pollen shells and SPPs are correlated.’

21. The findings of Hughes et al. (2020) should be considered and discussed in the context of the model evaluation. Key points to consider is the timing of SPP concentrations with respect to the arrival of the thunderstorm (e.g., line 289), their observation of SPP in the absence of lightning, concentrations of SPP observed relative to the model (e.g., line 447), their observations of higher SPP concentrations in strong storms with high windspeeds and downdrafts (which may help to explain differences across 20 and 21 November (e.g., line 479), and their observations of SPP in precipitation events of many types during tree pollen season (which suggests SPP events are quite common, but vary in strength; line 505-507).

The Hughes (2020) paper is methodologically interesting and shows pollen rupturing after most rain events, but it lacks any connection between the increased amounts of ruptured pollen in the air and a clinical outcome, such as an outbreak of acute bronchospasm and an increased demand on the hospital system. We’re not convinced that the findings of Hughes et al (2020) can explain the Melbourne event because there was very little rain. 

Line 57. ‘Hughes et al. (2020) were the first to determine SPPs under high rainfall conditions using single-particle fluorescence spectroscopy coincident with measurements of pollen markers such as fructose.’ 

At line 289 ‘ Key to the Hughes et al (2020) finding was that SPP increases were associated with high rates of rainfall coincident with thunderstorm activity. The difference in the Melbourne storm was the absence of rain. Lack of rain is the major factor in why so many people were outside at the time and were exposed.’ 

At line 369. ‘Marks et al. (23) did show increases in both pollen shells and whole grass pollen during a 1997 TA event in New South Wales, but the ratio of pollen shells to whole pollen did not increase due to the storm. By contrast, Hughes et al observed a decrease in whole pollen concentrations due to the rain.’

22. Interestingly, Hughes et al. (2020) estimate that at least one-third of the pollen grains ruptured in the convective storm highlighted in their article, similar to the fraction of total pollen grains that the pollen shells in this study, suggesting similarities in the number of pollens that rupture in the case of ryegrass in Melbourne.

Hughes et al (2020) say that at least one-fifth (or 20%) of the pollen in the air is ruptured, not one third. This is much less than the 70% rupturing rate observed by Taylor et al (2002) on ryegrass pollen in the laboratory. We’re using 70% as the fraction of pollen that ruptures, from Taylor et al (2002). 

23. The model described appears to allow for pollen rupturing at high humidity (section 3.1). Does the model consider and account for the closing of anthers at high humidity to minimize pollen exposure to humid air? This is a likely source of error contributing to the high estimates of SPP at night that is not consistent with observations.

The emission parameterisation for whole pollen at high humidity takes into account the closing of anthers (Emmerson et al., 2019). Thus whole pollen is only emitted when the humidity is low enough to do so. 

At line 361. ‘The condition of the 80% RH threshold required for pollen rupturing is also a condition that suppresses whole pollen emissions from the plant in VGPEM, via the closing of anthers at high humidity.’

24. In stating “we find that the RH throughout the atmosphere to be well below the 80% required for pollen altitude described in Taylor and Jonsson (2004), which is suggested to approach 100% at the cloud base? For example, in Figure 4, it appears that the 80% threshold is met by some models at a pressure of 600 hPa, which is relevant to pollen grains entrained in updrafts. The statement in the abstract that most pollen remained within 40 m of the surface at line 24 suggests that the model may not accurately represent the vertical distribution of pollen grains in the atmosphere.

The 80% threshold is only just met at Melbourne airport for the WRF model in figure 4. We show the spatial concentrations of SPPs in WRFC-CTM in figure S1, which shows that SPPs are present. However the key point of this paper is that more SPPs are produced regularly at other times, which does not make the mechanism a good one for the prediction of thunderstorm asthma (see line 508). 

At line 353 ‘ At the time of the storm in central Melbourne we find that the RH throughout the atmosphere to be well below the 80% required. The key finding is that more SPPs are produced regularly at other times when RH tends to be higher, such as at night rather than during a hot late spring day. This does not make the mechanism a good one for the prediction of thunderstorm asthma.’

We disagree that the model may not accurately represent the vertical distribution in the atmosphere. Observations by Damialis et al (2017) show that grass pollen is mainly found at the surface (>77% of total). Less than 18 % of total grass pollen was found at 2000 m.

Adjust line 333 ‘Modelled vertical concentrations of whole grass pollen are largely logarithmic, with most (62% average across models) situated in the first 40 m, similar to the ground-based measurements made by Damialis et al. (24) who showed 77% grass pollen at the surface and less than 18 % at 2000 m. It is only in the WRFC-CTM profile that the whole pollen may intersect with RH conditions being high enough for rupture, noting that observations plotted in Fig 4 were above Melbourne Airport whereas rupturing may have occurred at other locations in the model domain. ’

25. Like 396 – the statement that SPP are too small to contribute much to PM2.5 is not supported with any evidence. Meanwhile, prior work by Rathnayake et al. (2017) showed that pollen contributed 0.74 μg/m3 corresponding to 42% of PM2.5 mass on 2 May 2013 during tree pollen season when a thunderstorm struck.

Air quality monitoring sites in Melbourne reported a peak in PM10 at the time of the storm, but no coincident peak in PM2.5. Yet there was a thunderstorm asthma event. 

We will change line 396 to read “There was no increase in observed PM2.5 mass concentrations across Melbourne, although whole pollen and dust are generally found in more coarse size fractions. Assuming SPPs were responsible for the 2016 Melbourne event, the air quality observations suggest SPPs were too small to contribute much mass to PM2.5”.

26. In considering the role of lightning (paragraph beginning at line 433) – have the authors considered cloud-to-cloud lightning, or only cloud-to-ground lightning strikes? Both may be relevant to pollen rupturing, particularly to pollen grains at higher altitudes.

The WWLLN dataset includes both cloud-to-ground and cloud-to-cloud lightning. The Hughes et al (2020) paper suggests that neither cloud-to-cloud nor cloud-to-ground lightning was required for pollen fragments to occur. 

At line 239 “The WWLLN dataset includes both cloud-to-ground and cloud-to-cloud lightning, but preferentially detect the stronger cloud-to-ground occurences as noted by Virts et al. (25)”

27. Line 49 – Provide an explanation for how SPP would become concentrated at ground level.

We have added the gust front outflow to figure 1, as described in Marks et al (2001)

28. Figure 1 – Define the meaning of the symbols used, especially different types of arrows, plusses, and minuses.

+ and – signs represent the electrical charge within the thunderstorm cloud. We have added a key to remove some of the different arrow styles. 

29. Table 1 – It would be helpful to the reader if you defined the many variables in these equations as part of the table.

Frupt = fraction of whole pollen shells that rupture = 0.7. χ = airborne concentration of whole pollen, �g m-3, nspg = number of SPP per whole pollen grain = 700. Pfx = whole pollen emission rate, g m-3. Fpr = precipitation function. fRH relative humidity function. MSPP = mass of 1 SPP = 1.13 x 10-13 g. MPOL = mass of 1 whole pollen grain = 2.24 x 10-9 g. fWS = function of wind speed. WS = wind speed, m s-1. Nlightning = number of lightning occurrences.

30. Terminology – pollen “exine” is a more biologically accurate term than “shell”

Changed to exine

Works Cited

Hughes, D. D., C. B. A. Mampage, L. M. Jones, Z. Liu and E. A. Stone (2020). "Characterization of Atmospheric Pollen Fragments during Springtime Thunderstorms." Environmental Science & Technology Letters 7(6): 409-414.

Rathnayake, C. M., N. Metwali, T. Jayarathne, J. Kettler, Y. Huang, P. S. Thorne, P. T. O'Shaughnessy and E. A. Stone (2017). "Influence of rain on the abundance of bioaerosols in fine and coarse particles." Atmos. Chem. Phys. 17(3): 2459-2475.

Suphioglu, C., M. B. Singh, P. Taylor, R. Bellomo, P. Holmes, R. Puy and R. B. Knox (1992). "Mechanism of grass-pollen induced asthma." Lancet 339(8793): 569-572.

Taylor, P. E., R. C. Flagan, R. Valenta and M. M. Glovsky (2002). "Release of allergens as respirable aerosols: A link between grass pollen and asthma." Journal of Allergy and Clinical Immunology 109(1): 51-56.

Taylor, P. E. and H. Jonsson (2004). "Thunderstorm asthma." Current Allergy and Asthma Reports 4(5): 409-413.

References

1. Suphioglu C, Singh MB, Taylor P, Knox RB, Bellomo R, Holmes P, et al. Mechanism of grass-pollen-induced asthma. The Lancet. 1992 Mar 7;339(8793):569–72. 

2. Beggs H. GAMSSA – A New Global Australian Multi-Sensor SST Analysis, Submitted to Proceedings of the 9th GHRSST–PP Science Team Meeting, 9-13 June 2008. In Perros-Guirec, France; 2008. 

3. Lock AP, Brown AR, Bush MR, Martin GM, Smith RNB. A New Boundary Layer Mixing Scheme. Part I: Scheme Description and Single-Column Model Tests. Mon Wea Rev. 2000 Sep 1;128(9):3187–99. 

4. Edwards JM, McGregor JR, Bush MR, Bornemann FJ. Assessment of numerical weather forecasts against observations from Cardington: seasonal diurnal cycles of screen-level and surface temperatures and surface fluxes. Quarterly Journal of the Royal Meteorological Society. 2011;137(656):656–72. 

5. Hurley P. Modelling Mean and Turbulence Fields in the Dry Convective Boundary Layer with the Eddy-Diffusivity/Mass-Flux Approach. Boundary-Layer Meteorol. 2007 Dec 1;125(3):525–36. 

6. Janjić ZI. The Step-Mountain Eta Coordinate Model: Further Developments of the Convection, Viscous Sublayer, and Turbulence Closure Schemes. Monthly Weather Review. 1994 May 1;122(5):927–45. 

7. Wilson DR, Ballard SP. A microphysically based precipitation scheme for the UK meteorological office unified model. Quarterly Journal of the Royal Meteorological Society. 1999;125(557):1607–36. 

8. Rotstayn LD. A physically based scheme for the treatment of stratiform clouds and precipitation in large-scale models. I: Description and evaluation of the microphysical processes. Quarterly Journal of the Royal Meteorological Society. 1997;123(541):1227–82. 

9. Lin Y-L, Farley RD, Orville HD. Bulk Parameterization of the Snow Field in a Cloud Model. J Climate Appl Meteor. 1983 Jun 1;22(6):1065–92. 

10. Morrison H, Thompson G, Tatarskii V. Impact of Cloud Microphysics on the Development of Trailing Stratiform Precipitation in a Simulated Squall Line: Comparison of One- and Two-Moment Schemes. Monthly Weather Review. 2009 Mar 1;137(3):991–1007. 

11. Edwards JM, Slingo A. Studies with a flexible new radiation code. I: Choosing a configuration for a large-scale model. Quarterly Journal of the Royal Meteorological Society. 1996;122(531):689–719. 

12. Manners J, Thelen J-C, Petch J, Hill P, Edwards JM. Two fast radiative transfer methods to improve the temporal sampling of clouds in numerical weather prediction and climate models. Quarterly Journal of the Royal Meteorological Society. 2009;135(639):457–68. 

13. Schwarzkopf MD, Ramaswamy V. Radiative effects of CH4, N2O, halocarbons and the foreign-broadened H2O continuum: A GCM experiment. Journal of Geophysical Research: Atmospheres. 1999;104(D8):9467–88. 

14. Freidenreich SM, Ramaswamy V. A new multiple-band solar radiative parameterization for general circulation models. Journal of Geophysical Research: Atmospheres. 1999;104(D24):31389–409. 

15. Mlawer EJ, Taubman SJ, Brown PD, Iacono MJ, Clough SA. Radiative transfer for inhomogeneous atmospheres: RRTM, a validated correlated-k model for the longwave. Journal of Geophysical Research: Atmospheres. 1997;102(D14):16663–82. 

16. Essery R, Best M, Cox P. MOSES 2.2 Technical Documentation [Internet]. UK Meterological Office; 2001 [cited 2020 Nov 13]. Available from: https://www.yumpu.com/en/document/view/9128071/moses-22-technical-documentation-pdf-1-mb-met-office

17. Kowalczyk EA, Garratt JR, Krummel PB. Implementation of a soil-canopy scheme into the CSIRO GCM -- regional aspects of the model response. 1994 [cited 2020 Nov 13]; Available from: https://publications.csiro.au/rpr/pub?list=BRO&pid=procite:b6c301f7-9939-4df5-b67d-7cfbdcc0f632

18. Tewari M, Chen F, Wang W, Dudhia J, LeMone M, Mitchell K, et al. Implementation and verification of the unified NOAH land surface model in the WRF model. In: In: Paper 142A, 20th conference on Weather Analysis and Forecasting /16th conference on numerical weather prediction [Internet]. 2004 [cited 2020 Dec 7]. p. 6. Available from: https://ams.confex.com/ams/pdfpapers/69061.pdf

19. Gregory D, Rowntree PR. A Mass Flux Convection Scheme with Representation of Cloud Ensemble Characteristics and Stability-Dependent Closure. Mon Wea Rev. 1990 Jul 1;118(7):1483–506. 

20. Grell GA, Dévényi D. A generalized approach to parameterizing convection combining ensemble and data assimilation techniques. Geophysical Research Letters. 2002;29(14):38-1-38–4. 

21. Rotstayn LD, Lohmann U. Tropical Rainfall Trends and the Indirect Aerosol Effect. J Climate. 2002 Aug 1;15(15):2103–16. 

22. Wilson DR, Bushell AC, Kerr‐Munslow AM, Price JD, Morcrette CJ. PC2: A prognostic cloud fraction and condensation scheme. I: Scheme description. Quarterly Journal of the Royal Meteorological Society. 2008;134(637):2093–107. 

23. Marks G, Colquhoun J, Girgis S, Koski H, Treloar A, Hansen P, et al. Thunderstorm outflows preceding epidemics of asthma during spring and summer. Thorax. 2001;56(6):468–71. 

24. Damialis A, Kaimakamis E, Konoglou M, Akritidis I, Traidl-Hoffmann C, Gioulekas D. Estimating the abundance of airborne pollen and fungal spores at variable elevations using an aircraft: how high can they fly? Sci Rep. 2017 16;7:44535. 

25. Virts KS, Wallace JM, Hutchins ML, Holzworth RH. Highlights of a New Ground-Based, Hourly Global Lightning Climatology. Bulletin of the American Meteorological Society. 2013 Sep 1;94(9):1381–91.

---

## [Decision Letter · Decision Letter 1]

9 Feb 2021

PONE-D-20-27816R1

Atmospheric modelling of grass pollen rupturing mechanisms for thunderstorm asthma prediction.

PLOS ONE

Dear Dr. Emmerson,

Thank you for submitting your manuscript to PLOS ONE. After careful consideration, we feel that it has merit but does not fully meet PLOS ONE’s publication criteria as it currently stands. Therefore, we invite you to submit a revised version of the manuscript that addresses the points raised during the review process.

We note that Reviewer 3 has requested that you add several citations to your manuscript, and that you responded in the previous revision as to why you did not feel that these references were appropriate or relevant to your study. Reviewer 4 has agreed with your assessment. As such, please note that further consideration of your manuscript does not depend upon your inclusion of these citations.

Please focus your revisions on the comments provided by Reviewer 4, below.

We look forward to receiving your revised manuscript.

Kind regards,

Chon-Lin Lee, Ph.D.

Academic Editor

PLOS ONE

Reviewers' comments:

Reviewer's Responses to Questions

**Comments to the Author**

1. If the authors have adequately addressed your comments raised in a previous round of review and you feel that this manuscript is now acceptable for publication, you may indicate that here to bypass the “Comments to the Author” section, enter your conflict of interest statement in the “Confidential to Editor” section, and submit your "Accept" recommendation.

Reviewer #1: All comments have been addressed

Reviewer #2: All comments have been addressed

Reviewer #3: All comments have been addressed

Reviewer #4: (No Response)

2. Is the manuscript technically sound, and do the data support the conclusions?

Reviewer #1: Yes

Reviewer #2: (No Response)

Reviewer #3: Partly

Reviewer #4: Partly

3. Has the statistical analysis been performed appropriately and rigorously? 

Reviewer #1: I Don't Know

Reviewer #2: (No Response)

Reviewer #3: N/A

Reviewer #4: No

4. Have the authors made all data underlying the findings in their manuscript fully available?

Reviewer #1: Yes

Reviewer #2: (No Response)

Reviewer #3: Yes

Reviewer #4: No

5. Is the manuscript presented in an intelligible fashion and written in standard English?

Reviewer #1: Yes

Reviewer #2: (No Response)

Reviewer #3: Yes

Reviewer #4: Yes

6. Review Comments to the Author

Reviewer #1: The corrected version completely answers my questions and I consider this paper suitable for publication in PLOS ONE.

However, I have a problem with the "shell/exine" correction suggested by a fellow reviewer. The use of shell in order to refer to the empty pollen enveloppe was correct and I strongly disagree with the use of exine to replace shell. The term shell is used in microencapsulation study, see for example :

Diego–Taboada A, Beckett ST, Atkin SL, Mackenzie G (2014) Hollow pollen shells to enhance drug delivery. Pharmaceutics 6: 80–96

Chapter 24 - Pollen and Spore Shells—Nature’s Microcapsules, Microencapsulation in the Food Industry A Practical Implementation Guide 2014, Pages 283-297

The term exine refers to the outer layer of the pollen wall; it is only a part of the pollen enveloppe (Halbritter et al., Illustrated Pollen Terminology, 2018). I would suggest to use pollen shell, pollen enveloppe, empty (emptied ?) pollen, broken pollen grain, fragmented pollen, but not exine which make incomprehensible some sentences.

Minor comments.

L283 – the pollen mass should be 2.24 x 10-8 g not 2.24 x 10-9 g (22 ng)

Reviewer #2: Review Comments to the Author

I feel that the authors have addressed my concerns and the manuscript acceptable for publication.

Reviewer #3: Minor revision (but important for acceptance or rebuttal, see my enclosed comment).

I realize that authors refused to modify the manuscript following my suggestions . In particular they refuse , in my opinion without motivation, to add two references published in the two most important journals of allergology in the world (JACI and Allergy) . In my opinion these references have importance as a link between pathogenetic and clinical aspects of thunderstorm asthma to clarify the mechanism of rupture of pollen grains during a thunderstorm in pollen seasons and this point is important for the readers

Reviewer #4: This manuscript presents modeling studies of pollen grains and ruptured pollen (called pollen exine) surrounding the Melbourne, Australia thunderstorm asthma epidemic in 2016. The authors examine the extent to which current pollen rupturing mechanisms documented in the literature can parameterize the occurrence of sub-pollen particles. Overall, the authors reports show that the current understanding of pollen rupturing mechanisms (e.g., humidity > 80%, presence of lightning, wind speeds) are lacking and model predictions do not agree well with experimental observations. The authors conclude that a better understanding of pollen rupturing is needed to support modeling efforts and development of an early-warning system.

The authors have addressed most of the reviewer’s concerns. Additional improvements are needed prior to publication, which are outlined below. I concur with the authors that Reviewer 3’s comments appear are not applicable to the current manuscript.

1. The statistics reported are not yet to the high technical standard required by PLOS ONE. Correlations are widely used as a means of model comparison; however, only Pearson’s correlation coefficients (R values) are reported. When comparing measurement and model results, the slope of the relationship is also needed to compare the magnitude of the model prediction to the experimental data.

2. Line 167: Provide justification for the use of 0.7 for the fraction of whole grass pollen grains that rupture; i.e. that this is the fraction of ryegrass pollen grains that rupture in water after 5 minutes and is the only data available in the literature of its kind.

3. Clarification is needed in the discussion of observed and modeled pollen exines. Is it assumed in the model that one exine fragment is produced for one pollen grain?

4. In discussing Figure 2, please comment on the ability of the model to capture the magnitude of pollen exines. It appears that there is a large under-prediction in panel (b) and a large over-prediction in panel (c).

5. Also in discussing Figure 2 in the paragraph beginning at line 297, the authors should discuss the observations of pollen exine on 20 November. It appears that there is experimental evidence of high concentrations of ruptured pollen grains, although no epidemiological evidence of a thunderstorm asthma outbreak. This seems to suggest that pollen exine is not always linked to negative respiratory outcomes, and that it is not a good measure of allergenic sub-pollen particles that trigger thunderstorm asthma.

6. The discussion of the data from Damailis et al. (2017, doi:10.1038/srep44535) should be improved. In Damailis et al. (2017), Poaceae (grass) pollen had concentrations of 76 grains m-3 at 2000m (measured by aircraft) and 196 grains m-3 at the surface. This corresponds to a 2.6 times higher concentration at the surface compared to 2000 m. The numbers cited by the authors (in Damailis et al. Table 1) correspond to a the “contribution to the total concentration”, which represents the relative abundance of Poaceae pollen relative to other pollen types (17.6% at 2000m and 77.7% at ground level) which is not appropriate for model comparison in this context.

7. To resolve the error identified above, the authors should provide a comparable model comparison to these experimental data. This may be done by comparing the ratio of the modeled pollen grain concentration at the surface concentration to that at 2000 m. Observations show the surface is 2.6 times more concentrated than at 2000 m. A glance at Figure 4 panel d indicates the models predict this ratio to be in the range of 50 for WRF, 1000 CCAM, and 10,000 for ACCESS, suggesting a large under-estimation of pollen grain concentrations at 2000m and disagreement between the three meteorological models.

8. The authors should reconsider the accuracy of their vertical distributions of pollen grains as a likely source of model error. It appears like the model incorrectly predicts the vertical distribution of pollen grains, keeping them concentrated at the surface and away from higher RH at higher altitude where osmotic rupturing would occur. This is a concern raised by reviewer 4 (comment 24 in the response letter) that has not been adequately addressed in the revised manuscript, because the response incorrectly interprets the results of Damailis et al. (2017) as noted in the earlier comment. Relevant to this topic, are the broader results of Damailis et al. (2017) that airborne concentrations of pollen (at 2000m), measured by aircraft, exceeded those at ground level in the case of 12 of 14 observed pollen species, with Poaceae being one of two pollen types having higher concentrations at ground level. Overall, Damailis et al. (2017) conclude that “our data and the statistical analysis have actually proved lower [pollen and fungal spore] concentrations at lower elevations.”

9. Line 382 – explain why “this does not make the mechanism a good one for the prediction of thunderstorm asthma”, e.g., because it would lead to false positives.

10. PLOS ONE requires that all data be made available. It appears that only the radiosonde data is available (line 567), but the 24-h and hourly pollen and pollen exine counts are not listed. Please add these observational data to these data archives, or provide an alternate link to a permanent database.

Minor comments

11. Line 116: italicized word missing - pollen counts on either side of 21 November

12. Line 382: Cite the “rupturing experiments cited earlier” here, too, to avoid ambiguity.

13. Line 430: Please name the specific “air quality observations.” If not described in the methods of this paper, please provide a reference (e.g., to an online database of hourly PM2.5).

14. Line 460: Revise “when measurements of SPP concentrations are possible” following recent observations of SPP by Hughes et al. (2020, reference 18).

15. Clarify – SPP concentrations should not have units of “grains m-3” because they are not intact grains. A number concentration with units of “m-3” is more appropriate.

16. Line 523: Clarify – “As we did not determine the concentrations of SPPs….”

7. PLOS authors have the option to publish the peer review history of their article (what does this mean?). If published, this will include your full peer review and any attached files.

Reviewer #1: **Yes: **Nicolas VISEZ

Reviewer #2: No

Reviewer #3: **Yes: **Professor Gennaro D’Amato

Reviewer #4: No

---

## [Author Response · Author response to Decision Letter 1]

18 Feb 2021

Dear Editor,

As requested, we have focussed on responding to comments from reviewer #4. In particular we have completed another test to address their concerns about the vertical concentration of whole pollen in the models. By forcing higher concentrations of whole pollen at 2000 m in the model we have shown that rupturing still does not occur at the time of the storm due to low relative humidity. Higher rupturing rates remain at other times in the week, and mainly at night. The conclusion of the manuscript remains the same – that use of a relative humidity threshold to rupture pollen in a forecast system would lead to frequent false alarms of thunderstorm asthma.

We have also responded to the couple of comments from reviewer #1, and were happy to revert back to the nomenclature of pollen 'shells' at their request.

Please see the 'response to reviewers' file for full details.

Kind regards

Kathryn Emmerson

---

## [Editor Report · Decision Letter 2]

19 Mar 2021

Atmospheric modelling of grass pollen rupturing mechanisms for thunderstorm asthma prediction.

PONE-D-20-27816R2

Dear Dr. Emmerson,

We’re pleased to inform you that your manuscript has been judged scientifically suitable for publication and will be formally accepted for publication once it meets all outstanding technical requirements.

Kind regards,

Chon-Lin Lee, Ph.D.

Academic Editor

PLOS ONE
---

## [Editor Report · Acceptance letter]

24 Mar 2021

PONE-D-20-27816R2 

Atmospheric modelling of grass pollen rupturing mechanisms for thunderstorm asthma prediction 

Dear Dr. Emmerson:

I'm pleased to inform you that your manuscript has been deemed suitable for publication in PLOS ONE. Congratulations! Your manuscript is now with our production department. 

Kind regards, 

on behalf of

Dr. Chon-Lin Lee 

Academic Editor

PLOS ONE